# Technical Note: Flow velocity and discharge measurement in rivers using terrestrial and UAV imagery

Anette Eltner[1], Hannes Sardemann[1], Jens Grundmann[2]

[1]Institute of Photogrammetry and Remote Sensing, Technsiche Universität Dresden, Dresden, 01069, Germany
[2]Institute of Hydrology, Technsiche Universität Dresden, Dresden, 01069, Germany

*Correspondence to*: Anette Eltner (anette.eltner@tu-dresden.de)

**Abstract.** An automatic workflow to measure surface flow velocities in rivers is introduced, including a Python tool. The method is based on PTV and comprises an automatic definition of the search area for particles to track. Tracking is performed in the original images. Only the final tracks are geo-referenced, intersecting the image observations with water surface in object space. Detected particles and corresponding feature tracks are
filtered considering particle and flow characteristics to mitigate the impact of sun glare and outliers. The method can be applied to different perspectives, including terrestrial and aerial (i.e. UAV) imagery. To account for camera movements images can be co-registered in an automatic approach. In addition to velocity estimates, discharge is calculated using the surface velocities and wetted cross-section derived from surface models computed with structure-from-motion and multi-media photogrammetry. The workflow is tested at two river reaches (paved and natural) in Germany. Reference data is provided by ADCP measurements. At the paved river reach highest deviations of flow velocity and discharge reach
4 % and 5 %, respectively. At the natural river highest deviations are larger (up to 31 %) due to the irregular cross-section shapes hindering accurate contrasting of ADCP- and image-based results. The provided tool enables the measurement of surface flow velocities independently of the perspective from which images are acquired. With the contact-less measurement spatially distributed velocity fields can be estimated and river discharge in previously ungauged and unmeasured regions can be calculated, solely requiring some scaling information.

## 1 Introduction

Measuring discharge of rivers is a major task in hydrometry because of its importance in many hydrological and geomorphological research questions, e.g. to understand the characteristics of catchments and their adaption to climatic changes. Different approaches exist to apply the velocity-area-method to measure discharge relying on information about the flow velocity and the wetted river cross-section area. Established tools to retrieve flow velocities are the application of current meters, acoustic devices (i.e. acoustic Doppler current profilers) or surface velocity radar

(Herschy, 2008, Merz, 2010, Morgenschweis, 2010, Gravelle, 2015, Welber et al. 2016). However, these velocity estimation methods are either labour intense, require minimum water depths, need prolonged measurement periods or can endanger the operator during flood measurements.

A promising alternative are remote sensing tools utilising image-based approaches. Due to their flexibility (only a camera is needed) they are used frequently exploiting various sensors and platforms for data acquisition. For instance, RGB sensors have been used (e.g. Muste et al., 2008) as well as thermal cameras (e.g. Puelo et al., 2012). Ran et al. (2016) demonstrated the suitability of a low-cost Raspberry Pi camera to observe flash floods and Le Coz et al. (2016) and Guillén et al. (2017) illustrated the usability of crowd-sourced imagery for post-flood analysis. Image-based setups allow for the assessment of temporally changing flow dynamics (Sidorchuk et al., 2008) due to the potential continuous recording of entire river reaches. Furthermore, small-scale investigations are enabled as shown by Legout et al. (2012), who measured the spatial distribution of surface runoff from mm- to cm-depth, at a range where other methods are failing.

Various algorithms exist for surface flow velocity monitoring from image-based observations deploying tracking tools. Four tracking approaches are applied frequently in the field to monitor rivers. The first method is large scale particle image velocimetry (LSPIV) originally introduced by Fujita et al. (1998). This approach uses tracking of features at the water surface that are caused due to natural occurring floating particles or free surface deformations caused by ripples or waves e.g. due to wind or turbulences (Muste et al., 2008). In general, the area of interest (i.e. the water surface) is divided in sub-regions and these sub-regions are used as templates. In the subsequent images, the corresponding areas are searched for using correlation techniques.

Fujita et al. (2007) advanced the LSPIV approach by an algorithm called space time image velocimetry (STIV). STIV performs faster, because tracking is performed in 1D instead of 2D. Profiles are extracted along the main flow direction to subsequently draw particle movements along the time axis (i.e. change along the profiles within succeeding frames) leading to a space-time image. The resulting angle of the pattern within that image resolves into the flow velocity.

The third possibility is the usage of optical flow algorithms developed in the computer vision community. For instance, the Lucas-Kanade (Lucas & Kanade, 1981) operation has been utilized to measure surface velocities of large floods or small rivers (Perks et al., 2016 or Lin et al., 2019, respectively). The method aims to minimize grey scale value differences between template and search area adapting the parameters of an affine transformation within an optimization procedure. Finally, particle tracking velocimetry (PTV) is a tracking option that uses correlation techniques as in LSPIV. However, instead of using entire sub-regions as templates single particles are detected first and then searched for in the subsequent images.

LSPIV is the most widely used method and can be considered as matured (Muste et al., 2011). Amongst others, it enabled the measurement of the hysteresis phenomena during flood events (Tsubaki et al., 2011, Muste et al., 2011). However, LSPIV mostly underestimates velocities, which is

revealed in more detail by Tauro et al. (2017), who prefer PTV instead. In contrast to LSPIV PTV does not assume similar flow conditions for the entire search area and it is not influenced by surface frictional resistance (Lewis and Rhoads, 2015) or standing waves (Tsubaki et al., 2011).

Besides surface flow velocity another parameter has to be considered to derive discharge measurements from image-based tracking approaches. The depth averaged flow velocity, used in the velocity-area-method, does not necessarily correspond to the surface flow velocity, which is amongst others due to the influence of river bed roughness. Therefore, a so called velocity coefficient has to be used to adjust the surface velocities (Creutin et al., 2003, Le Coz et al., 2010). Usually, the deeper the flow the higher the coefficient is assumed (Le Coz et al., 2010). The coefficient can vary with different river cross-sections (Le Coz et al., 2010) and it can change within the same cross-section due to varying water depths, which is likely for irregular profiles (Gunawan et al., 2012). Muste et al. (2008) state that the coefficient mostly ranges between 0.79 to 0.93, but values as low as 0.55 have been measured (Genc et al., 2015). Considering the correct velocity coefficient is important because it has a high impact on the discharge estimation error in remote sensing approaches (Dramais et al., 2011).

When flow-velocities and velocity coefficient are known, the area of the river cross-section is needed to calculate the discharge with the velocity-area method (e.g. Hauet et al., 2008). Different tools exist for contactless river cross-section area measurement. Muste et al. (2014) show that it is possible to use velocity pattern measured with LSPIV to retrieve flow depth in shallow flow conditions. Another approach is the utilization of ground penetrating radar as illustrated for larger rivers by Costa et al. (2000). An additional increasingly used method to retrieve the topographic (and thus cross-section) information of the river reach is the usage of structure-from-motion (SfM) photogrammetry (Eltner et al., 2016). For instance, Ran et al. (2016) capture stereo images to reconstruct the 3D information of a river reach from overlapping images during low flow conditions. However, if water is present during data acquisition and the river bed is still visible the underwater measurements have to be corrected for refraction impacts (Mulsow et al., 2018) or else heights of points below the water surface will be underestimated. Woodget et al. (2015) introduce a workflow to account for refraction using a constant correction value for the case of Nadir viewing image collection. Dietrich (2017) extends this correction procedure for the case of oblique imagery. Detert et al. (2017) were the first to perform fully contact-less, image-based discharge estimations using refraction corrected river cross-sections (adapting Woodget et al., 2015) and surface flow velocities, all measured from UAV imagery. However, the authors relied on a seeded flow to apply LSPIV.

Image-based tracking approaches can be applied to imagery captured terrestrially as well as from aerial platforms. In the case of aerial imagery, the utilization of UAVs for data acquisition is increasing. The advantage of drones is their flexibility and allowance to capture runoff patterns during high flow conditions (e.g. Tauro et al., 2016, Perks et al., 2016, Detert et al., 2017, Koutalakis et al., 2019), even enabling real-time data processing (Thumser et al., 2018). However, a challenge to overcome is the correction of camera movements during the UAV flight. Although camera mounts are commonly stabilised, remaining motions need to be accounted for. Tauro et al. (2016) subtract velocities measured in stable areas outside the river from velocities tracked in the river. Another possibility is the usage of co-registration. Thereby, either features (e.g. SIFT features; Lowe,

2004) are searched for in stable areas (Fujita et al., 2015, Blois et al., 2016) or Ground Control Points (GCPs) are detected (Le Boursicaud et al.,

2016). Subsequently, these image points are matched across the images. Afterwards, this information is used to apply a perspective transformation to each image to fit them to a reference image. However, so far stable areas are still masked manually.

In the case of terrestrial data acquisition, the conversion of pixel measurements to metric velocity values is more challenging compared to UAV data due to a stronger deviation of the perspective from an orthogonal projection, which leads to decreasing accuracies with increasing distance to the sensor. Therefore, Kim et al. (2008) suggest to avoid camera setups with tilting angles larger than 10°. Most approaches ortho-rectify the images

prior tracking to allow for a correct scaling of the image tracks. However, performing the tracking in the original image would be favoured to minimize interpolation errors, especially for oblique camera setups, and to solely transform the tracked image point coordinates into object space (Stumpf et al., 2015).

Several software tools already exist to perform image-based velocimetry (e.g. PTVlab from Brevis et al. 2011, PIVlab from Thielicke and Stamhuis 2014, Fudaa-LSPIV at https://forge.irstea.fr/projects/fudaa-lspiv, KU-STIV developed by Fujita, or RIVeR from Patalano et al. 2017). These tools

cover different processing steps and tracking options to retrieve surface flow velocities and discharge. In this study, we combine the entire workflow from video, either captured with UAV or from terrestrial camera, to velocity of river reaches, considering image stabilization, automatic feature search area extraction, PTV, track filtering and metric velocity retrieval via forward ray intersection. An automatic flow velocity measurement tool (FlowVelo tool) for image velocimetry is presented and provided by public domain to overcome existing gaps discussed before. It is independent from the data acquisition scheme and relies on PTV. Camera movements are accounted for in a fully automatic approach if a sufficient amount of

shore area is visible. Furthermore, the search area for features to track, i.e. the river area, is extracted automatically solely requiring water level information and a 3D surface model of the river reach. The 3D surface model is calculated from image data with SfM photogrammetry additionally considering multi-media photogrammetry to retrieve both, topography and bathymetry. In order to improve tracking results, detected features and velocity tracks are filtered with different methods. Finally, we estimate discharge from surface flow velocities and cross sectional areas. The FlowVelo tool and the whole workflow are investigated for two river reaches, paved and natural, at which velocities and discharges are compared

to ADCP references.

## 2 Methods

In this study, the FlowVelo tool is introduced that allows for the measurement of flow velocity from videos independently from the acquisition platform, i.e. either aerial or terrestrial. Different parameter options for feature detection and tracking as well as track filtering are explained. Two

experimental study sites have been chosen to evaluate the performance of video based flow velocity estimation using camera frames acquired from
different perspectives. First the experimental study sites are introduced and afterwards the tool is explained.

## 2.1 Areas of interest

The experimental study sites are short river reaches in Saxony, Germany (fig. 1). One studied river reach is situated at the Wesenitz. This river originates in the Lausitzer highlands, has a catchment size of about 280 km², and exhibits a river length of 83 km. The area of interest is located at the river gauge station Elbersdorf, which is operated by the Saxon state company for environment and agriculture. Here, annual average water level
and discharge for the hydrological year 2017 are 48 cm and 2.4 m²/s, respectively. Field campaigns were conducted on March 31st and on April 4th 2017. During the campaigns water level amounted 51 cm (discharge 2.7 m³/s). The investigated river section at the Wesenitz is paved but influenced by local sand banks at the river bottom. During the data acquisition the river had a width of about 10 m.

The other river is the Freiberger Mulde, which originates in the Ore Mountains, has a catchment size of about 2980 km², and displays a river length of 124 km. The area of interest is located close the gauge Nossen. Average discharge and water level for the hydrological year 2016 are 5.6 m³/s
and 65 cm, respectively. The gauge station is located 1 km upstream of the studied river reach. The field campaign was conducted on October, 26th 2016. During this day discharge and water level were 5.7 m³/s and 68 cm. The approximated river width was 15 m. The chosen region of interest at the Freiberger Mulde is a natural river section with non-uniform flow conditions.

## 2.2 Data acquisition

Different data was collected during the field campaigns at both river sections. Amongst others ADCP measurements were performed as flow
velocity reference, GCPs were defined to geo-reference the video data and UAV and terrestrial imagery were acquired to perform image-based flow velocity estimation.

### 2.2.1 ADCP measurements

For the ADCP measurements the moving boat approach with StreamPro from RDI is used. Velocity profiles were measured with a blanking range of 14 cm near the water surface. Data were processed using the AGILA software from the German Federal Institute of Hydrology (BfG).
Measurements along the boat track were projected onto a reference cross sectional area. Afterwards surface flow velocities were extrapolated to allow for a comparison to the image-based values. For the extrapolation, power functions were fitted to the measured vertical velocity profile for each individual ADCP ensemble using the software AGILA (for more detail see Adler, 1993 and Morgenschweis, 2010). Then, velocities at the water surface were calculated with these functions. Thus, all ADCP measurements of the profile were considered to extrapolate surface velocities.

At the Wesenitz ADCP measurements were performed at one cross-section in eight repetitions (fig. 1a). Average water surface velocity was about

0.7 m/s and resulting discharge amounts 2.7 m³/s (table 1). At the Freiberger Mulde three cross-sections were chosen (fig. 1b) to acquire data that allows for a spatially distributed assessment of the image-based data. Average river surface velocities ranged between 0.60 m/s and 0.76 m/s (table 1).

The spatial variation of flow velocities is larger at the Freiberger Mulde, where measurements were performed in a natural river reach, which is in contrast to the flow velocity range at the Wesenitz, where data was captured at a standardized gauge station. Thus, only one profile was measured

at the Wesenitz. The discharge at the Freiberger Mulde is 5.88 m³/s on average but reveals a standard deviation of 0.25 m³/s. Estimated discharge of the river reach therefore reveals a variation of about 4 %, which can be attributed to inconsistencies during data acquisition. A decrease of the velocity coefficient, which has been derived from the ADCP measurements, with decreasing water depth is observed. At the Freiberger Mulde cross-sections 1 and 3 (Table 1) have lower water depth compared to profile 2. Thus, the velocity coefficients are lower.

### 2.2.2 Image-based data

At both river reaches video sequences were acquired with terrestrial cameras and with a camera installed at the UAV Asctec Falcon 8. The airborne image data was captured at flying heights of about 20 m and 30 m at the Wesenitz and Freiberger Mulde, respectively. Videos were captured with a frame rate of 25 frames per second (fps) and with a resolution of 1920 x 1080 pixels using the camera Sony NEX-5N with a fixed lens with a focal length of 16 mm. The ground sampling distance (GSD) is about 7 mm at the Wesenitz and about 9 mm at the Freiberger Mulde.

The terrestrial cameras were installed at bridges across the river (fig. 1). At the Wesenitz three cameras were installed to evaluate the performance

of different cameras (fig. 1a). Two Canon EOS 1200D and one Canon EOS 500D were setup. The 1200D cameras captured video sequences with 25 fps and with a resolution of 1920 x 1080 pixels. The 500D captured frames with a higher rate (30 fps) and smaller image resolution (1280 x 720 pixels). All three cameras were facing downstream. At the Freiberger Mulde the camera Casio EX-F1, equipped with a zoom lens fixed to 7.5 mm, was used. Videos were captured with 30 fps and a resolution of 640 x 480 pixels. The camera was facing upstream.

The terrestrial cameras were calibrated for both rivers to allow for the correction of image distortion impacts. To estimate the interior geometry of

the cameras, images of an in-house calibration field have been captured in a specific calibration pattern (Luhmann et al., 2014). These images, together with approximate coordinates of the calibration field and approximations of the interior camera orientations were used in a free-network bundle adjustment within Aicon 3D Studio to calibrate each camera. More details regarding the workflow are given in Eltner and Schneider (2015).

### 2.3 High resolution topography of the river reaches

Local 3D surface models describing the topography of the river reaches are necessary to scale the image measurements. Therefore, high resolution topography data was acquired at both rivers using SfM photogrammetry (Eltner et al., 2016, James et al., 2019). SfM in combination with multi-view stereo matching (MVS) allows for the digital reconstruction of the topography from overlapping images and some GCPs. Thereby, homologous image points in overlapping images are detected and matched automatically. From these homologous points and some assumptions about the interior camera model, the position and orientation of each captured image (i.e. camera pose) can be calculated. With known network geometry, a dense point cloud can be computed, reconstructing the 3D information for almost each image pixel. The resulting 3D surface models are geo-referenced during the reconstruction or afterwards via GCPs.

At the Wesenitz the 3D surface model of the river reach was calculated from 85 terrestrially captured images with a Canon EOS 600D (20 mm fixe lens) and from 20 UAV images (Eltner et al., 2018). The SfM calculations were performed in Agisoft Metashape. At the Freiberger Mulde seven frames of the video sequence, which is also used for later PTV processing, were utilized to perform SfM photogrammetry to retrieve the corresponding 3D model of the river reach.

GCPs made of white circles on a black background were installed in order to reference the 3D data as well as the image-based velocity measurements. They were measured with a total station at the Freiberger Mulde and during the first campaign at the Wesenitz. During the second campaign at the Wesenitz GCPs were extracted from cobblestone corners (with sufficient contrast) at the gauge, which are visible in the terrestrial images used for the 3D model reconstruction. GCPs were measured in at least five images for sufficient redundancy and thus more reliable coordinate calculation.

The bathymetric information of the river reaches was retrieved using the same UAV data as for the topographic information above the water level. Refraction impacts are accounted for using the tool provided by Dietrich (2017). Underwater points, camera poses and interior camera parameters as well as the water level need to be provided. The corrected point clouds can be noisy and were therefore filtered and smoothed in CloudCompare using a statistical outlier filter to detect isolated points and using a moving least square filter. Eltner et al. (2018) revealed that cm-accuracies can be reached using multi-media SfM at the Wesenitz river reach. Due to opaque water conditions at the Freiberger Mulde, imagery of a previous UAV flight (six weeks earlier and with no flood events happening during that period) had to be used to reconstruct the underwater area.

### 2.4 Surface Flow Velocity Workflow – the FlowVelo tool

This chapter introduces the general approach to measure surface flow velocities from either terrestrial or airborne video sequences. Thereby, essential processing steps are described in more detail. The FlowVelo tool is realized in Python and using the OpenCV library (Bradski, 2000). Fig. 2 illustrates the entire data processing workflow of the tool.

### 2.4.1 Frame preparation

Video sequences are converted into individual frames prior to the data processing. Afterwards, image co-registration is necessary if the camera is not stable during video capturing, as it is the case for the UAV data. Each frame of the entire video sequence is co-registered to the first frame of the same sequence to correct camera movements and thus to enable that all frames capture the same scene. This processing step is preformed fully automatically. In each frame Harris corner features are detected (Harris & Stephens, 1988), which are then matched to the first frame of the sequence using SIFT (Lowe, 2004) or ORB descriptors (Rublee et al., 2011). The suitability of co-registration in different conditions and over longer periods of time has been illustrated in Eltner et al. (2018), who introduce a terrestrial camera gauge for water level measurements.

Harris features in the water region are detected as outliers due to their changing appearance between subsequent frames leading to matching failure. And if moving features in the water area still might be matched, they are latest filtered during the parameter estimation of the homography because these points will be considered as outliers during the model fitting with RANSAC (Fischler & Bolles, 1981). Thus, only stable and reliable homologous image points outside the river are kept and used to calculate the homography parameters between the first frame and all subsequent frames. Finally, a perspective transformation is applied to ensure that all frames fit to the first image. It has to be mentioned that this approach is only working as long as enough stable areas are visible on both river shores.

In the FlowVelo tool five parameters can be set to adjust the co-registration of each individual scenery. The maximum number of keypoints defines how many features are maximally searched for in each frame. Larger numbers can increase the robustness and accuracy of matching but also the processing time. The number of good matches determines how many matched features between two frames are needed minimally to find the homography. Again, larger values increase the robustness, but they can also lead to a failure of processing if fewer feature matches are found than appointed here. Furthermore, it can be defined, which feature descriptor is chosen for matching, if features are matched back and forth increasing the accuracy and processing time, and if image co-registration is performed to the first frame or in a series to each consequent frame of the sequence.

### 2.4.2 Finding features to track

A search area in the river region has to be defined to detect particles before tracking. This is due to the circumstance that most feature detectors look for regions with high contrast. Therefore, points of interest would be found on the land, where contrast is usually higher than on the water surface. Thus, in a first step the river area has to be masked in the images and defined as the search area for tracking before applying particle detection.

**Feature search area and pose estimation**

The feature search area is a region of interest that is defined as a function of the water level to mask the image. The water level and a 3D surface model of the river reach (fig. 3a) observed by the camera have to be known to define this water area automatically. The 3D surface model is clipped

with the water level value to keep solely the points below the water surface. Afterwards, these points are projected into image space (fig. 3b). Therefore, information about the pose and the interior geometry of the camera is necessary. In the FlowVelo tool information about the camera pose is either estimated with spatial resection considering the GCP coordinates in image and object space and the interior camera parameters (for more details see Eltner et al., 2018) or it can be simply stated if the pose has been defined by other measures.

Next, the 3D point cloud of the observed river reach is projected into a 2D image (fig. 3b). To fill gaps, potentially arising for 3D surface models with low resolution, a morphological closing is performed. Finally, the contour of the underwater area is extracted to define the search mask for the individual frames. If several contours are detected, the largest contour is chosen. If a 3D surface model is not present for automatic feature search area detection, the area of interest for tracking can also be provided via a mask file.

**Feature detection and filtering**

Particles are detected with the Shi-Tomasi feature (or good feature to track; GFTT) detector (Shi & Tomasi, 1994). Thereby, features are detected similar to the Harris corner detector but a different score is considered to decide for a valid feature (fig. 3c). Many more feature detectors are possible. Tauro et al. (2018) test several methods and show that the GFTT detector performs well and also finds features in regions of poor contrast. The elimination of particles, which are not suitable for tracking, is necessary. For instance, reflections of sunlight at waves showing high contrasts on the water surface need to be removed to avoid erroneous tracking of fake particles (Lewis and Rhoads, 2015). Therefore, a nearest neighbour search is performed to find areas with strong clusters of particles. If there are too many features within a defined search radius, the particle will be excluded from further analysis. In addition, features are removed that reveal brightness values below a threshold, e.g. to avoid the inclusion of wave shadows as features.

### 2.4.3 Feature tracking

When features have been detected, they are tracked through subsequent frames (fig. 4). This tracking is performed using normalised cross correlation (NCC). Normalization allows accounting for brightness and illumination differences between different frames. The positions of the detected features are chosen to define templates with a specific kernel size (mostly 10 pixels in this study, Appendix 2). In the next frame NCC is performed within a defined search area (mostly 15 pixels in this study, Appendix 2) to find the positions with highest correlation scores for each feature, potentially corresponding to the new positions on the water surface of the migrated particles.

To refine the matching, an additional subpixel accurate processing is performed. Thereby, template and matched search area of the same size are converted into the frequency domain to measure the phase shift between both and afterwards the subpixel peak location is determined with a weighted centroid fit. The final matched locations define the new templates for tracking in the next frame. This tracking approach is performed for a specified number of frames. In this study, features are tracked for 20 frames and new features are detected every 15[th] frame. It can be suitable to

detect features more frequently than the number of frames they are tracked across, because features can change their appearance and new features
can enter the area of view although the already detected features are still tracked.

### 2.4.4 Track filtering

Figure 4 shows that false tracking results can still occur, e.g. tracks that significantly deviate from the main flow direction. This is amongst others due to remaining speckle detected as features or due to tracking of features with low contrast leading to ambiguous matching scores. Therefore, resulting velocity tracks need to be filtered. Tauro et al. (2018) remove false trajectories considering minimum track length and track orientation.

In this study, we also make assumptions about the flow characteristics of the river (fig. 5). We consider six parameters; minimum frame amount of a tracked feature, minimum and maximum tracking distances, flow steadiness, range of track directions, and deviation from average flow direction. Each track has to fulfil these criteria to be considered as a reliable velocity information. Thereby, each track is the combination of the individual sub-tracks from frame to frame, with feature detection performed in the first frame.

The first criterion considers the minimum percentage of frames across which the features have to be traceable (here 65 % ).The underlying

assumption is that if the feature is only traceable across a few frames then it is more likely not a well-defined flowing particle at the water surface but may for instance be a speckle occurrence due to sun glare. However, the minimum value can be set to 0 to avoid any constraints regarding flow velocities and camera frame rates.

The second and third filter criteria are the distances across which features were tracked, comprising thresholds for minimum and maximum distances. The distance thresholds can be roughly approximated when image scale and the range of expected river flow velocity is known. In this

study, the minimum and maximum distance parameters are set to 0.1 and 10 pixels, respectively.

The fourth criterion considers the directional flow behaviour of the feature with a steadiness parameter. Therefore, directions of sub-tracks (from frame to frame) are analysed for each track. Tracks are excluded when the standard deviation is above a defined threshold (30° in this study). The idea is that river observations are performed during nearly uniform flow conditions. Thus, high frequencies of changes in flow directions within a track indicate measurement errors and should be filtered. In addition to this steadiness parameter, the range of all sub-track directions is also

considered as a measure of the flow behaviour. If the range is above a defined threshold, the track will be excluded (here 120°).

For the last criterion the main flow direction of the river is examined. The average direction of all tracks is calculated and if the direction of the individual tracks are larger or below a buffer threshold (here 30°), they are rejected from further processing. The buffer value has to be defined considering the general variability of the river surface flow pattern. The lower the parameter is chosen the more a uniform flow is assumed. It has to be noted that the directional filter has a limited applicability in more complex flow conditions, e.g. turbulent, non-uniform rivers. In such situation,

local filters should be preferred over these global values.

### 2.4.5 Velocity retrieval

In the last processing stage, measured distances are transformed from pixel values to metric units to receive flow velocities in the unit of m/s. With known camera pose and interior camera geometry image measurements can be projected into object space. This leads to a 3D representation of the light ray emerging from the image plane and proceeding through the camera's projection centre. 3D object coordinates of an image measurement can be calculated by intersecting its ray with a 3D surface model of the river. In this case the water surface, assumed as planar at the water level, defines the location of intersection. The starting and ending points of each track are intersected with the water plane to retrieve real world coordinates. From the distance between start and end, and considering the camera's frame rate as well as the number of tracked frames, metric flow velocities are retrieved. Finally, the metric velocity tracks are filtered once more with a statistical outlier filter to remove remaining outliers (fig. 6). The threshold is defined as the sum of the average velocity with a multiple of its standard deviation (e.g. Thielicke & Stamhuis, 2014). The lower the multiple is chosen, the more features will be filtered and only tracks will be kept, which have values close to the average velocity. In this study, the parameter was set to 1.5. This processing step is more important for challenging tracking situations.

Regarding tracking reliability, it should be noted that in the case of terrestrial cameras with an oblique view onto the river velocity measurements are preferred closer to the sensor. Particles move across a larger number of pixels in close range to the camera than in further distances, e.g. an erroneous measurement of 1 pixel close to the camera might result to measurement error of 1 cm whereas in further distance it can correspond to 1 m. Furthermore, tracking accuracy decreases significantly in far ranges due to increasing glancing ray intersections with the water surface.

### 2.5 Discharge estimation

The bathymetric information as well as the flow velocities are needed to calculate the discharge. Thereby, sole UAV data can be used as shown by Detert et al. (2017). In this study, we cut river cross-sections from the reconstructed bathymetry and topography at the approximate locations of the ADCP measurements. Afterwards, we extract the water level information by manually detecting the water line in at least three overlapping images and spatially intersecting these point measurements in the object space.

The surface flow velocity values are averaged and multiplied with the velocity coefficients estimated from the ADCP measurements to account for depth-averaged velocities (Table 1). This approach is suitable at the Wesenitz. But at the Freiberger Mulde the method is restricted due to the irregular river cross-sections limiting the application of a constant velocity coefficient. Finally, discharge is estimated by multiplying the cross-section area with the depth-averaged velocity.

## 3 Results and Discussion

In this chapter, results of the accuracies of the image processing are displayed, tracked flow velocities are evaluated, and discharge estimations are analysed.

### 3.1 Accuracy assessment of camera pose estimation and image co-registration

To enable an accurate measurement of flow velocities it is necessary to consider how well the camera pose has been estimated. Furthermore, for cameras in motion the accuracy of frame co-registration has to be evaluated as well, to ensure that tracked movements of the particles indeed correspond to river flow instead of camera movements.

The accuracy of camera pose estimation can be estimated because more than three GCPs are available. In general, the camera pose will be calculated more accurately if GCPs are distributed around the area of interest in the object space and if images capture them in such a way that they cover the entire image extent because it allows for a stable image-to-object geometry. Furthermore, for highest accuracy demands GCPs need to be measured with high accuracy in object space and ideally with sub-pixel accuracy in image space. At both river reaches accuracies are better for the terrestrial cameras (table 2), which is due to a higher GSD as cameras are significantly closer to the area of interest compared to the UAV cameras. At the Wesenitz, another reason for the larger deviations is the circumstance that well marked, artificial GCPs were used for the terrestrial images, whereas GCPs were extracted from the 3D surface models to estimate the UAV camera pose leading to lower point coordinate accuracies.

Small template regions (10 pixels in size) in stable areas have been chosen (fig. 6) to estimate the accuracy of frame co-registration. At the Freiberger Mulde only GCPs could be used as templates because the remaining area of interest is covered by vegetation that changes frequently. At the Wesenitz cobble stone corners close to the river surface are chosen because it is important to see how well co-registration performs close to the water body for which velocities are estimated. Each extracted reference location is tracked through the frame sequence via NCC. In case of a perfect alignment, the templates should remain at the same image location throughout the sequence. In this study, at the Freiberger Mulde average deviation between tracked frames to the first frame amounts $0.5 \pm 0.6$ pixels for all templates, which corresponds to a co-registration accuracy of $4.3 \pm 5.2$ mm. At the Wesenitz, co-registration reveals an accuracy of $1.0 \pm 1.6$ pixels ($6.8 \pm 11.3$ mm). The lower image coverage of the right shore at the Wesenitz leads to a lower quality of the frame co-registration when compared to the Freiberger Mulde reach because features for frame matching are only kept outside the water area as the appearance of the river surface changes too quickly. Therefore, higher deviations are measured at the right shore than at the left shore. Considering only the matched targets at the left river side reveals an error range similar to the Freiberger Mulde.

## 3.2 Flow velocity measurements at the Wesenitz

The tracking results and retrieved flow velocities show a diverse picture for the different cameras. For instance, the final number of flow velocity tracks is different for each device (table 2). The lowest number of tracks is measured for the UAV camera. However, this camera solely captured a very short video sequence (about three seconds) that could be used for tracking. Furthermore, GSD of the UAV data is much lower than the GSD of the terrestrial cameras due to a larger sensor to object distance. The terrestrial cameras reveal a significantly denser field of flow velocity tracks (fig. 7). The terrestrial cameras captured videos of a length of about half a minute. Although video lengths of the terrestrial cameras are similar, the

number of final velocity tracks is varying. The camera closest to the water surface and with the least oblique view (1200D-II) reveals the highest track number. Camera 1200D-I reveals a lower number of velocity measurements, although frame resolution and focal length are the same and video length is even longer. The third camera (500D) depicts lowest track number, which is mainly due to a lower frame resolution.

Besides considerations of the camera geometry, track filtering is another very important aspect to retrieve reliable velocity measurements. The filtered track number is about a magnitude lower than the raw track amount for the terrestrial cameras (table 2) highlighting the importance of video

sequences with sufficient temporal duration. Thus, tracking should be performed as long as possible to increase the robustness of velocity filtering.

Comparing the range of flow velocity values between the different terrestrial cameras and the UAV camera reveals a good fit (fig. 7), which also coincides with the ADCP reference (table 3). Furthermore, regions of faster and slower velocities are revealed in the terrestrial image data that also show within the acoustic data. The average deviation of all cameras to the ADCP measurements are calculated for video-based track values that are within a maximal perpendicular distance to the ADCP profile of 1 m. The difference amounts to $0.03 \pm 0.06$ m/s. However, it is difficult

to perform exact comparison to the ADCP measurements because the precise location of the ADCP cross-section in the local coordinate system of the river reach is not known as the ADCP boat was not equipped with any positioning tool and its movement across the water surface was neither tracked nor synchronised. Therefore, accuracy assessment of the spatial velocity pattern is limited. Nevertheless, we were able to identify the start and end points of the cross-sections at the shore in the imagery. Therefore, we could approximately estimate the locations of the cross-sections in the decimetre range, which allows for velocity comparison if the surface flow velocity pattern does not become too variable within shortest

distances. This has to be kept in mind, when assessing the velocity differences, especially at the Freiberger Mulde.

Average surface flow velocities from the image-based measurements are higher or similar to the (extrapolated) ADCP retrieved surface velocity of 0.7 m/s, except for camera 1200D-II, which depicts lower values; also compared to the other cameras (table 4). A potential reason is the different coverage of the cross-section with measured velocity values. 1200D-II reveals the highest velocity value density and covers a larger part of the cross-section (fig. 7). More regions with lower velocities are measured by the 1200D-II, whereas the other cameras feature less cross-section

coverage and more values are measured in areas of faster velocities.

Interestingly, an impact of the missing camera calibration of the UAV images is not obvious. Lens distortion parameters were only modelled for the terrestrial cameras but were discarded for the UAV camera. The impact is assumed to be minimal because the camera distortion is usually especially large for cameras with very wide angles, which is not the case for the UAV camera. Furthermore, the distortion impact is more important when features are tracked for large distances in the image, which is also not the case in this study because features are mostly tracked between subsequent frames for only a few pixels.

### 3.3 Flow velocity measurements at the Freiberger Mulde

At the Freiberger Mulde a more diverse spatial velocity pattern becomes obvious (fig. 8). Especially the UAV data reveals areas of increased and decreased velocities along the river reach. Velocity ranges coincide with the ADCP measurements. Average deviations of the closest tracks to the reference values (similar approach to chapter 3.2) are on average $-0.01 \pm 0.07$ m/s for the terrestrial and UAV camera and for all cross-sections. However, velocities are either overestimated or underestimated at different profiles and for different cameras (table 3). The flow velocities for the UAV data is lower at profile 3 compared to the reference. However, due to the strong changes of flow velocities within short distances, especially at cross-section 3 (fig. 1), a possible reason can be false mapping of ADCP values to image-based values. The assumption that velocity underestimation at that cross-section is due to imprecise point-based velocity comparison is backed when comparing the average cross-section UAV-retrieved surface velocity (table 4) with the average ADCP velocity. In that case, the UAV data reveals larger values (0.79 versus 0.76 m/s, respectively), confirming the observations at cross-section 1 and 2.

The terrestrial camera depicts a lower spatial density of velocities compared to the terrestrial cameras at the Wesenitz (table 2), although the video sequence has comparable length. This is due to the significantly lower image resolution as well as the larger distance to the object. Therefore, less features are detectable. The average flow velocity at cross-section 1 as well as the average of contrasted individual velocity tracks are smaller than the reference. However, error behaviour of the image-based data might be less favourable at cross-section 1, where the comparison is made for image tracks measured at the far reach of the image. The sharp glancing angles at the water surface lead to higher uncertainties of the corresponding 3D coordinate.

The decision about how to set the parameters for tracking (e.g. patch size) and filtering (e.g. statistical threshold) remains challenging, especially in long-term applications when spatio-temporal flow conditions can change strongly (Hauet et al., 2008). Thus, in future studies intelligent decision approaches for corresponding parameters need to be developed, for instance where measurements are performed iteratively with changing parameters.

### 3.4 Camera based discharge retrieval

Discharge estimations at the Wesenitz do not show large deviations between the cameras because velocity estimates showed low deviations, as well (table 4). Solely camera 1200D-II displays a lower discharge. Average discharge for all cameras amounts to 2.7 m³/s, which corresponds to the discharge measured by the ADCP. Deviations to the reference are below 4 %, highlighting the great potential of UAV application to retrieve

discharge estimates solely from image data in regular river cross-sections. Standard deviations of the discharge estimations due to the consideration of the standard deviation of the surface flow velocities is small, ranging from 0.18 m³/s (7 %) to 0.56 m³/s (8 %) at the Wesenitz and Freiberger Mulde, respectively (table 4).

At the Freiberger Mulde, discharge estimates do not fit as well to the reference measurements. Velocities are only observed in the main flow of the river, where flow velocities are higher. Deviations to the ADCP reference are larger for the terrestrial camera, whose measurements are only

compared to profile 1, which shows a large range of flow velocity and depicts very low values outside the main flow (fig. 1). Comparing single velocity values to nearby ADCP measurements, instead of comparing averaged cross-section information, reveals that the accuracies of image-based velocity measurements are indeed higher (table 3). Neglecting the slower flow velocities in the shallower river region outside the main flow leads to overestimated discharge values for the irregular shaped cross-sections, which is in contrast to the regular cross-section at the Wesenitz. In addition, using the average velocity coefficient is adverse because the irregular profile shape indicates a changing coefficient (Kim et al., 2008).

Another important issue that needs to be noted is the circumstance that the image-based discharge estimation reveals a high variability that is sensitive to the defined wetted cross-section extracted with the defined water level. For instance, at the Wesenitz already 1 cm offset in the water level value causes a discharge difference of 0.08 m³/s (3 %) and 3 cm cause a difference of 7 % (0.2 m³/s). Different studies already highlight that the correct water level is important for accurate discharge estimation due to the wetted cross-section area error but that it is less relevant for the accuracy of the flow velocities due to erroneous ortho-rectification (Dramais et al., 2011, Le Boursicaud et al., 2016, Leitao et al., 2018).

### 3.5. Limits and perspectives

In this study, a workflow for surface flow velocity and discharge measurements in rivers using terrestrial and UAV imagery was tested successfully. In general, three main processing steps are necessary, i.e. retrieving terrain information via SfM photogrammetry, estimating the flow velocity with PTV and eventually calculating the discharge with the information from both previous steps. However, some constraints need to be considered. The FlowVelo tool requires at least the video frames, the camera pose (either estimated within in the tool considering GCP information or provided

externally), the water level and some estimates of the interior camera geometry (at least focal length and sensor size and resolution are needed). Furthermore, if the camera was not stable during the image acquisition, camera movements can be corrected automatically if sufficient shore areas are visible in the frames. With this information and pre-processing scaled river surface velocities are retrievable fully automatically.

However, some characteristics of the tool have to be considered. One aspect is the shore visibility in the frames for the co-registration. To guarantee at larger rivers stable areas that are large enough increasing the flying height might be necessary, potentially reducing the visibility of features to

track. Alternatively, cameras with wider opening angles might be needed, potentially resulting in stronger lens distortions. Furthermore, assumptions about the flow characteristics need to be made for successful filtering, which implies either some experience with image velocimetry in riverine environments or some trials to find the most suitable filtering parameters. Consideration of a suitable choice of the threshold of the statistical outlier filter is important, as well. If the filter is chosen too strictly, it can lead to the loss of valid velocity tracks, which is especially probable in rivers with complex flow patterns and a large range of velocities. Another important factor of the image velocimetry tool to consider is

the impact of the choices of thresholds on processing time. On the one hand, the more often features are detected and the more frames they are tracked across, the more reliable and robust tracking results are possible because track filtering will receive a larger sample for processing. However, tracking more features across an increased number of frames also increases processing time significantly, which is especially relevant for cameras with high frame rates and image resolutions. Nevertheless, in this study the maximum processing time (for the terrestrial cameras at the Wesenitz that captured videos with lengths of about half a minute) was still below 5 minutes on an average computer.

Measuring surface velocities implies sensitivities to external impacts such as winds, waves, or raindrops, potentially falsifying an already established ratio between surface and average flow velocity, i.e. velocity coefficient, due to decreasing or increasing the surface velocity depending on the wind and wave direction and velocity. However, windy and rainy conditions should be avoided using any surface velocity measurement. The accuracy and reliability of the surface velocity measurement can be improved by adding traceable particles to increase the seeding density as shown by Detert et al. 2017. In this study, only natural particles floating at the river surfaces at both study areas were used, which did not cover the entire observed

cross-section, leading to data gaps complicating the retrieval of discharge from the sparsely distributed velocity values.

The FlowVelo tool does not provide discharge information, yet, because discharge estimation requires additional parameters, which need to be determined prior using image velocimetry as an accurate automatic remote sensing approach. For instance, the water level and the related cross sectional area are needed as well as the velocity coefficient has to be known, which is a point of uncertainty especially at irregular river reaches. In this study, the velocity coefficient was estimated from the ADCP measurements dividing the mean velocity of the cross-section with the average

surface velocity. However, alternative approaches, e.g. hydraulic modelling, should be analysed in more detail in future studies to evaluate if they can support the retrieval of more suitable velocity coefficients. This becomes especially interesting due to novel possibilities of high resolution bathymetric and topographic data, e.g. using SfM approaches for river mapping.

**Conclusion**

In this study, we introduce a remote sensing workflow for automatic flow velocity calculation and discharge estimation. The approach can be
applied to terrestrial as well as aerial imagery. Thus, the importance of the acquisition scheme is secondary. However, visibility of tracked particles across the entire river cross-section is relevant as indicated by comparison of three different terrestrial cameras observing nearly the same river reach but revealing variations in the velocity estimates.

Camera movements during the video acquisition are stabilized using an automatic image co-registration method. To estimate flow velocity, particles on the water surface are detected and tracked using PTV. A feature search area is defined automatically solely relying on information about the
water level and the topography of the river reach. The detected and tracked particles are filtered with cluster analysis and by making assumptions about the flow characteristics. Discharge is retrieved using the depth-averaged flow velocity and the wetted cross-section, which is derived from a 3D surface model reconstructed with multi-media photogrammetry applied to UAV imagery.

Two study sites have been observed with different terrestrial cameras and with a UAV platform. Comparing the results with ADCP reference measurements reveal a high accuracy potential for surface flow velocities calculated with PTV and automatic image co-registration, especially at
standard gauging setups (maximal error of 4 %). At irregular cross-sections accuracy assessment of velocity tracking is limited due to high demands of position accuracies of the reference measurements. Discharge estimates with maximal errors of 5 % could be achieved at the standard track cross-section. At irregular profiles discharge calculation reveals significantly higher differences to reference measurements of 7 - 31 %. This is, amongst other reasons, due to incomplete velocity measurements across the entire river cross-section, leading to discharge overestimation when tracks are only retrieved in the faster flowing river region. Thus, further improvements of the tool for irregular cross-sections as well as considering
artificial flow seeding is advisable in future studies.

The workflow, including the provided velocity tracking tool FlowVelo tool, allows for a contact-less measurement of spatially distributed surface velocity fields and to estimate river discharge in previously ungauged and unmeasured regions, making it especially suitable for applications to assess flood events.


*Code and data availability*. The data used in this study and the tracking tool FlowVeloTool are available at http://dx.doi.org/10.25532/OPARA-32 and https://github.com/AnetteEltner/FlowVeloTool, respectively.

*Author contributions*. AE conceptualized the study, wrote the Python tool and drafted the manuscript. AE, HS, JG acquired, processed and analysed the data. HS and JG reviewed the draft.

*Competing interests*. The authors declare no competing interests.

*Acknowledgements*. We thank the European Social Fund (ESF) for funding this project (grants 100270097). These investigations are part of the research project "extreme events in small and medium catchments (EXTRUSO)." Furthermore, we are grateful for provided data sources by Andreas Kaiser and the Saxon state company for environment and agriculture. And we thank André Kutscher for helpful input to the tracking toolbox. The data used in this study and the tracking software FlowVeloTool are available at OPARA and https://github.com/AnetteEltner/FlowVeloTool, respectively. Finally, we are grateful for the reviews provided by Salvatore Manfreda and one anonymous referee that helped to improve the original
manuscript.

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

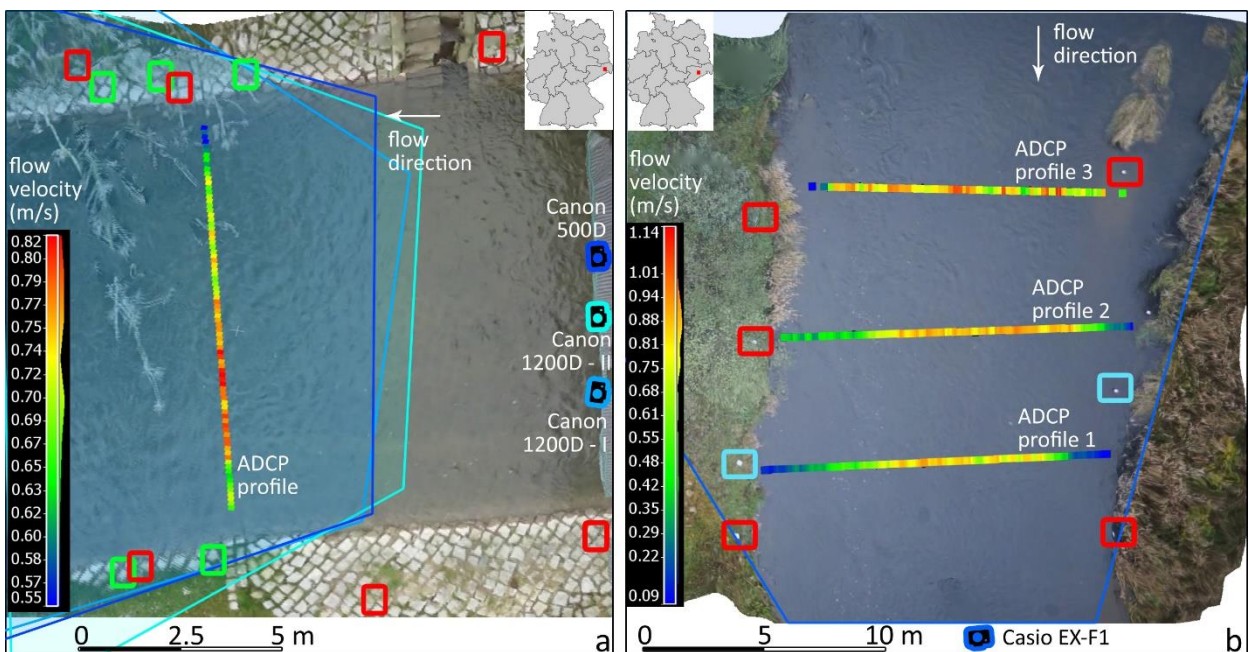

Figure 1: Areas of interest at the Wesenitz (a) and the Freiberger Mulde (b) displayed with UAV orthophotos calculated from video frames. Surface flow velocities measured with an ADCP and the corresponding locations of the measurement cross-sections within the river are illustrated. Ground control points (GCPs) are used to reference the image data at both river reaches. Red squares highlight GCPs used for terrestrial and UAV data at the Freiberger Mulde and UAV data at the Wesenitz. Green squares show location of GCPs used for terrestrial imagery at the Wesenitz. Check points (blue squares) are used to assess the accuracy of the 3D reconstruction from video frames at the Freiberger Mulde. Camera locations of the

terrestrial image sequence acquisition are illustrated as pictograms and corresponding image extent areas are shown (displayed area of interests in RGB correspond to the aerial image extents).

Table 1: River velocities measured with ADCP

| | Profile | Mean velocity (m/s) | Mean surface velocity (m/s) | Max. surface velocity (m/s) | Velocity coefficient (-) | Cross-section area (m²) | Discharge (m²/s) |
|---|---|---|---|---|---|---|---|
| Wesenitz | - | 0.59 | 0.70 | 0.82 | 0.84 | 4.63 | 2.72 |
| Freiberger Mulde | 1 | 0.48 | 0.60 | 0.92 | 0.80 | 11.75 | 5.60 |
| | 2 | 0.58 | 0.70 | 0.93 | 0.83 | 10.45 | 6.01 |
| | 3 | 0.59 | 0.76 | 1.03 | 0.78 | 10.30 | 6.04 |

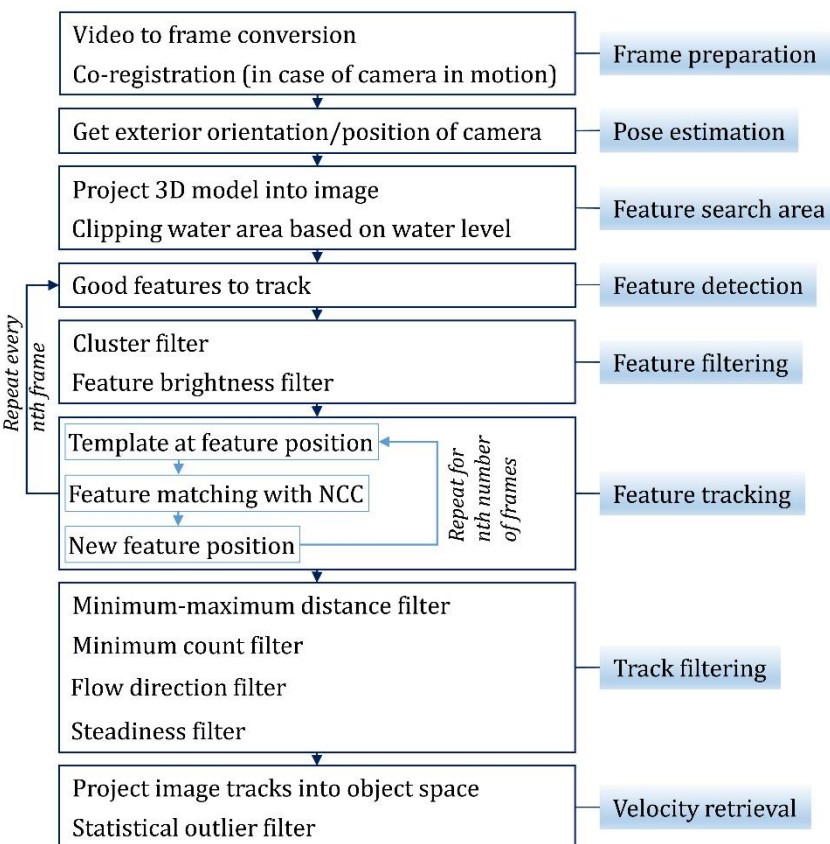

Figure 2: Workflow to retrieve flow velocities from video sequences.

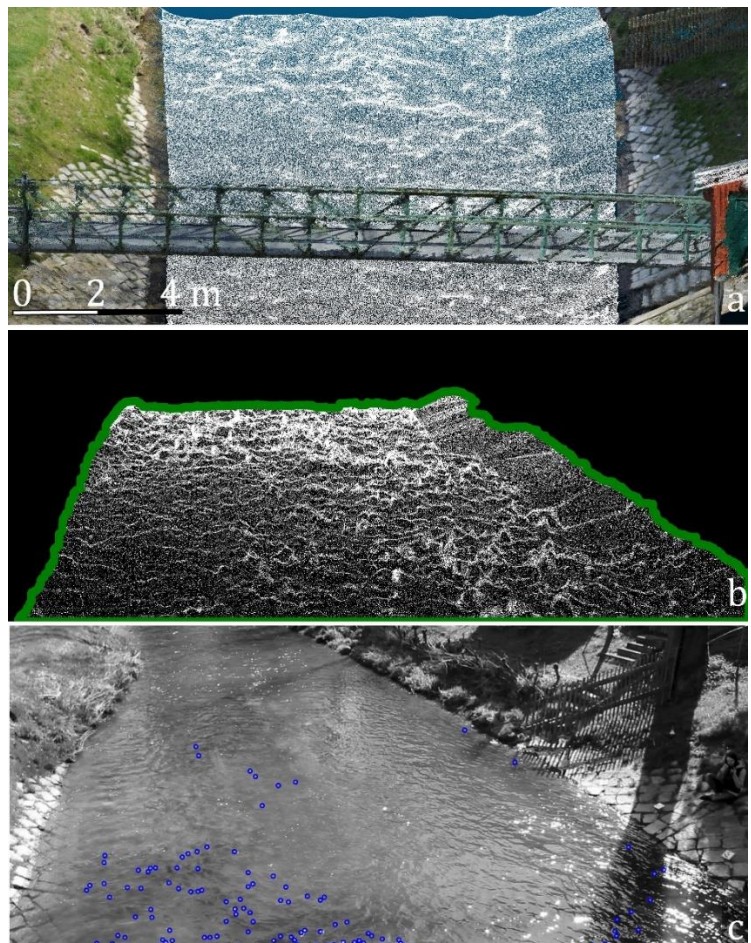


Figure 3: Defining the search area to extract particles to track. a) 3D point cloud of the investigated river reach at the Wesenitz. Coloured points (colorized with RGB information according to their real world object colour) are 3D points above the water surface reconstructed with SfM photogrammetry. White points are 3D points below the water surface reconstructed with SfM and corrected for refraction effects. b) 3D point cloud below the water level projected into image space. Green line depicts contour line, which is used as search mask for feature detection. c) Detected

and filtered features considered for tracking (blue circles).

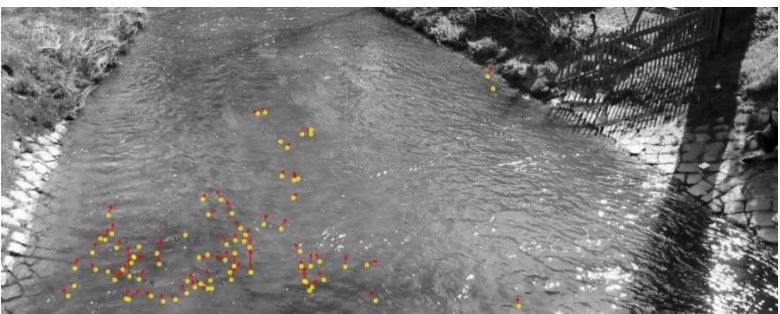

Figure 4: Exemplary display of the tracking result from one frame to the next.


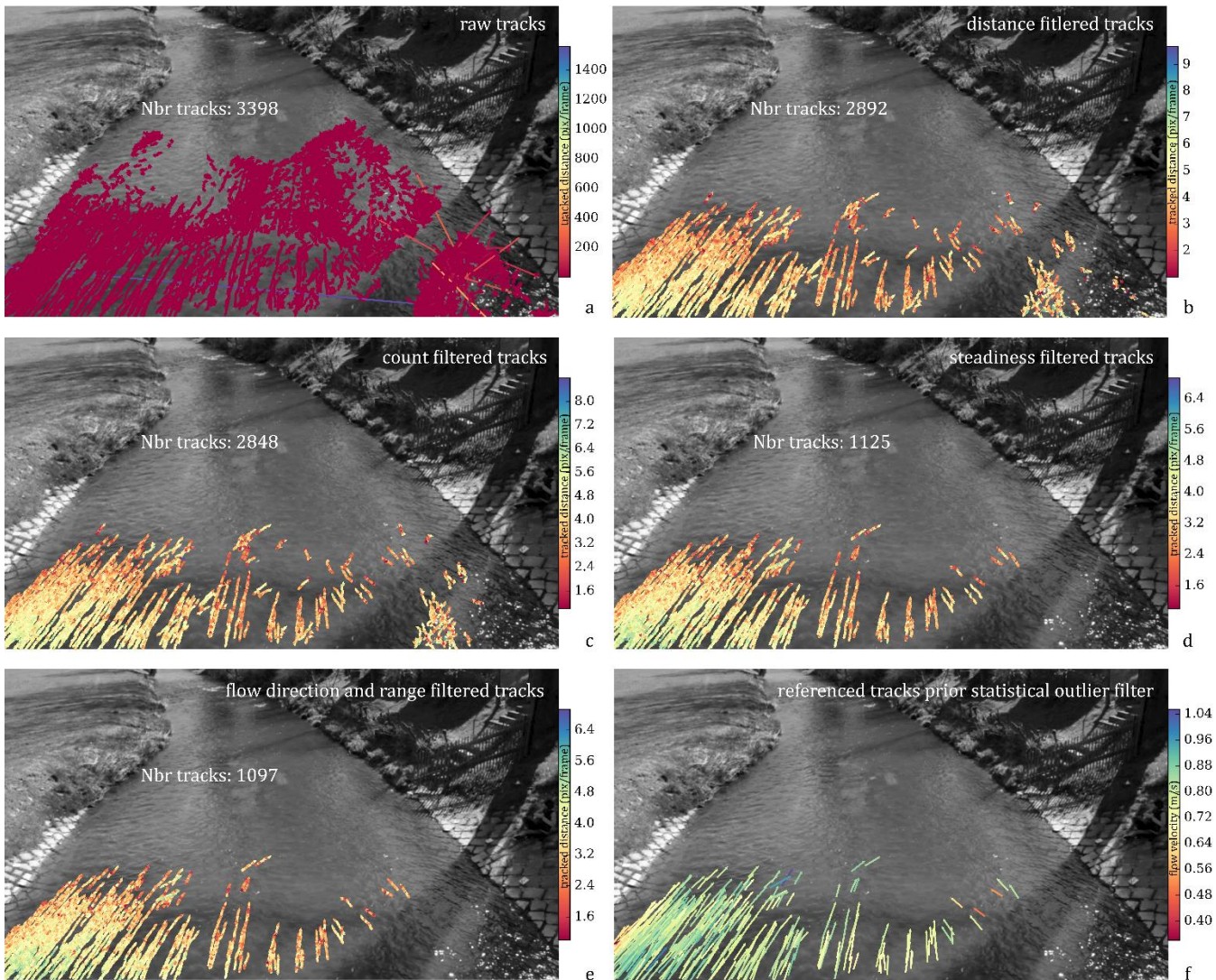

Figure 5: Result of tracked features after filtering has been applied to the video sequence of camera 500D (with a temporal length of 23 seconds). Sub-tracks are displayed and the number of tracks refers to full tracks (combination of sub-tracks). a) Raw tracks prior any filtering. b) Filtered tracks after applying minimum and maximum distance thresholds. c) Filtered tracks after applying a minimum count of sub-tracks over which features need to be tracked. d) Filtered tracks after considering standard deviation of sub-track directions. e) Filtered tracks after considering

deviation from average flow direction and range of orientation angels of sub-tracks. f) Filtered tracks converted into metric values to receive flow velocities in the unit m/s.


Table 2: Accuracy of camera pose estimation and density of tracking results. s0 corresponds to the average reprojection error after the adjusted spatial resection.

| | | accuracy | | | s0 (pixel) | number of frames | tracking density | |
| | | standard deviation (m) | | | | | number of raw tracks | number of final tracks |
| | | X | Y | Z | | | | |
| Wesenitz | UAV camera | 0.172 | 0.274 | 0.162 | 1.1 | 78 | 271 | 58 |
| | 500D | 0.027 | 0.066 | 0.039 | 0.9 | 690 | 3552 | 439 |
| | 1200D-I | 0.042 | 0.169 | 0.080 | 3.2 | 700 | 4781 | 603 |
| | 1200D-II | 0.041 | 0.127 | 0.073 | 1.6 | 640 | 14786 | 1239 |
| Frei. Muld. | UAV camera | 0.085 | 0.078 | 0.031 | 0.5 | 73 | 844 | 126 |
| | Casio EX-F1 | 0.018 | 0.010 | 0.015 | 0.3 | 750 | 3886 | 334 |


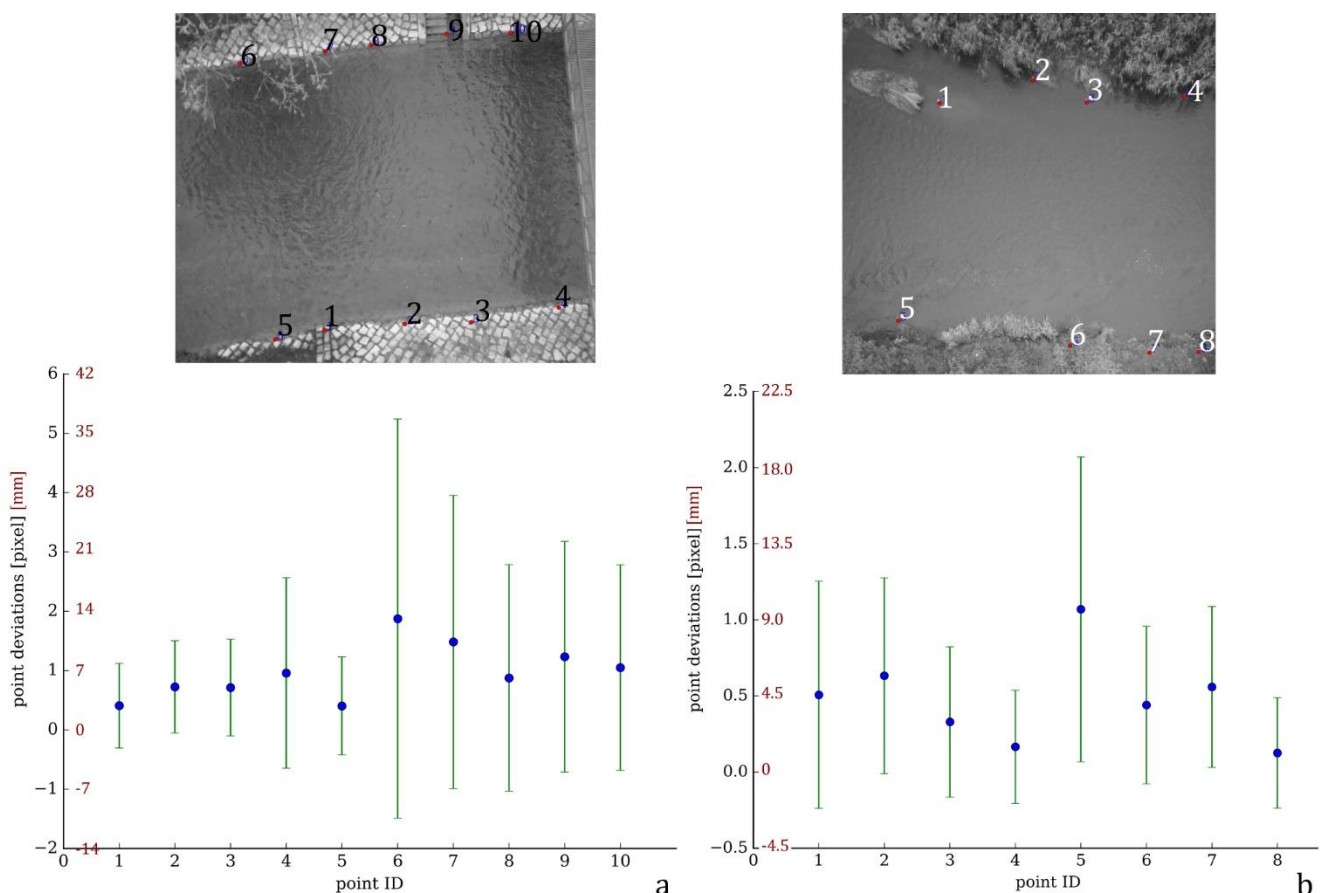

Figure 6: Accuracy of co-registration of video frames to single master frame displayed in image space (black axis) and the corresponding accuracy in object space (red axis) at the river Wesenitz (a) and Freiberger Mulde (b). Note that the images are only extracts form the original (bigger) images.


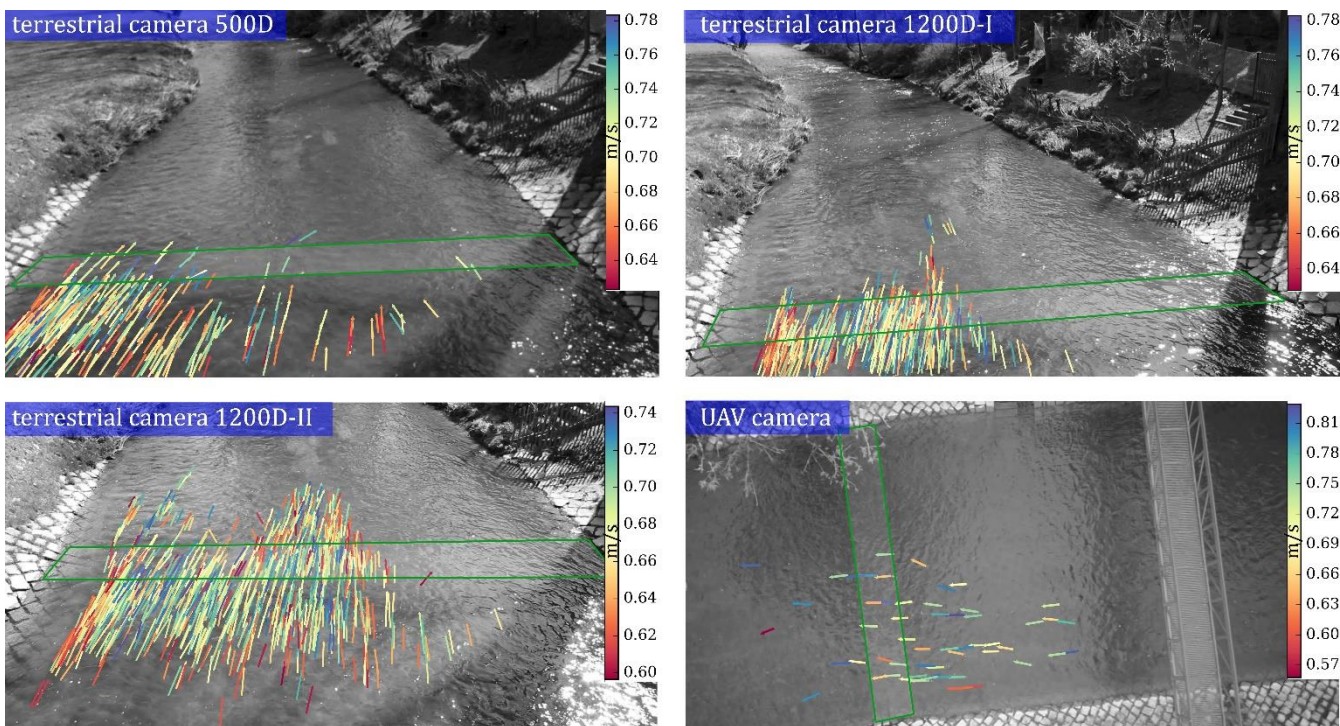

Figure 7: Flow velocities estimated at the Wesenitz using video frames captured with three different terrestrial cameras and a camera on an UAV platform. Final tracks after statistical outlier filter (fig. 5f) are displayed. Green border indicates area, in which image-based measurements are compared to ADCP velocities.


Table 3: Deviation between ADCP measurements and video based flow velocities. Differences are calculated for tracks within a range of 1 m and closest to the ADCP measurements.

| | | | surface velocity difference [m/s] | | |
| | | | average | standard deviation | track count |
|---|---|---|---|---|---|
| Wesenitz | UAV camera | | 0.03 | 0.07 | 10 |
| | 500D | | 0.00 | 0.06 | 24 |
| | 1200D-I | | 0.02 | 0.07 | 56 |
| | 1200D-II | | 0.08 | 0.06 | 88 |
| | average | | 0.03 | 0.06 | - |
| Freiberger Mulde | UAV camera | profile 1 | 0.03 | 0.09 | 8 |
| | UAV camera | profile 2 | 0.01 | 0.06 | 8 |
| | UAV camera | profile 3 | -0.05 | 0.06 | 8 |
| | Casio EX-F1 | profile 1 | -0.01 | 0.05 | 10 |
| | average | | 0.01 | 0.07 | - |


Table 4: Discharge estimated using flow velocities and cross-sections retrieved from UAV data.

| | | | average surface flow velocity at the cross-section [m/s] | Cross-section area [m²] | Discharge[m³/s] | |
| --- | --- | --- | --- | --- | --- | --- |
| | | | | | Average | Standard deviation |
| Wesenitz | UAV camera | | 0.71 | | 2.73 | 0.27 |
| | 500D | | 0.72 | 4.57 | 2.76 | 0.15 |
| | 1200D-I | | 0.71 | | 2.72 | 0.16 |
| | 1200D-II | | 0.67 | | 2.58 | 0.16 |
| | average | | | | 2.70 | 0.18 |
| | std dev | | | | 0.08 | |
| Freiberger Mulde | UAV camera | profile 1 | 0.79 | 11.61 | 7.34 | 0.74 |
| | | profile 2 | 0.77 | 10.35 | 6.60 | 0.60 |
| | | profile 3 | 0.79 | 9.19 | 5.64 | 0.43 |
| | Casio EX-F1 | profile 1 | 0.76 | 11.61 | 7.06 | 0.46 |
| | average | | | | 6.66 | 0.56 |
| | std dev | | | | 0.75 | |


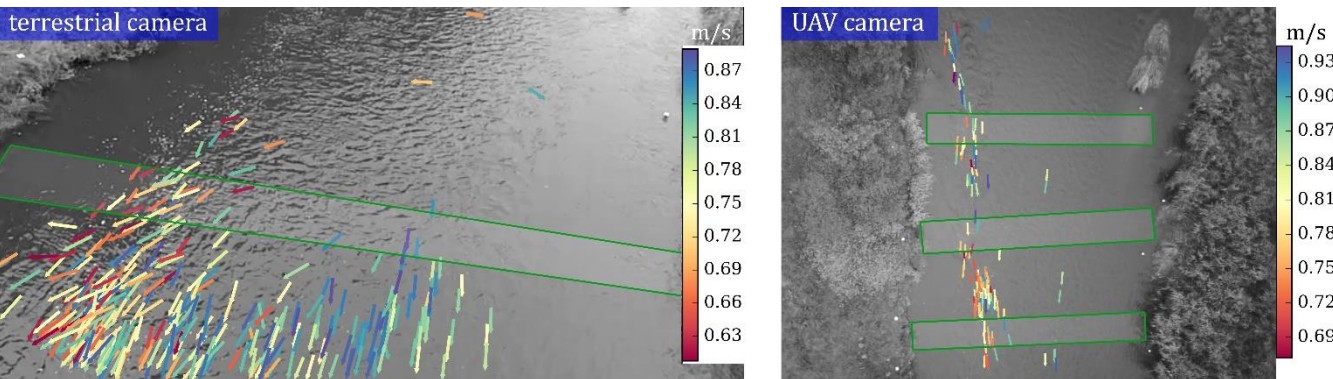

Figure 8: Flow velocities estimated at the Freiberger Mulder using video frames captured with a terrestrial camera and a camera on an UAV platform. Green border indicates area, in which image-based measurements are compared to ADCP velocities.
