# Peer review of "Technical Note: Flow velocity and discharge measurement in rivers using terrestrial and UAV imagery"

_Hydrology and Earth System Sciences, 2019_

## Referee Comment (RC1) · Salvatore Manfreda (Referee) · 19 Jul 2019

The main contribution of the present manuscript is oriented in exploring surface flow velocities and discharge estimations using fixed cameras and UASs devices. A full and automatic workflow is introduced for the estimation of the variables mentioned above. Two case studies are considered for validation purposes, namely the Wisenitz (paved) and Freiberger Mulde (natural). ADCP data were collected for benchmarking purposes. The manuscript is almost well written and easily understandable. Its length is also appropriate.

Major comments:

• Section 2.2.1 Reference data: The authors stated that surface flow velocities were

extrapolated using ADCP measurements. However, they do not say anything about the process. Please, add information on the extrapolating process. Additionally, at Wesenitz case study, only one cross-section was measured. Why such a decision? (consider that for a rigours comparison between image-velocimetry results and reference velocities is better not to use only local reference velocities).

• Section 2.2.2 Image-based data: The authors used a low-resolution camera for video acquisitions at the Freiburger Mulde case study. Justify such a decision considering that low-cost smartphones can reach a better resolution.

• Section 2.4.3 Feature tracking: The authors stated, 'In this study, features are tracked for 20 frames and new features are detected every 15th frame'. Is there any reason for these numbers? Why did the authors decide a new detection every 15 frames?

• Section 2.4.4 Track filtering: This subsection is relevant and deserves a better explanation of the filtering criteria. For example, it would be positive to add a figure showing an example of application of the different filtering criteria (e.g. what is the reference for the angles?).

• Section 2.4.5 Velocity retrieval: The authors stated: 'The threshold is defined as the sum of the average velocity with a multiple of its standard deviation'. Please, add information about the 'multiplying factor' of the standard deviation.

• Section 2.4.5 Velocity retrieval: The authors stated: 'For a better visualisation, final flow velocity tracks are rasterized'. Please, add information about the block assumed for the rasterizing process. If the comparison of estimated velocities is made with the rasterized velocities, please mention it and discuss implications.

• Section 3.2 Flow velocity measurements at the Wesenitz: The authors stated 'However, it is difficult to perform exact comparison to the ADCP measurements because the precise location of the ADCP cross-section in the local coordinate system of

the river reach is not known as the ADCP boat was not equipped with any positioning tool and its movement across the water surface was neither tracked nor synchronised. Therefore, the accuracy assessment of the spatial velocity pattern is limited'. This is a critical issue that may limit the validation of the procedure. Do you have any alternative strategy to quantify ADCP positions in order to allow a realistic comparison?

Minor comments:

⇢ Page 2, Line 2: '...observe flash floods. And Le Coz et al...'. Please, remove the point before 'and'. ⇢ Page 2, Line 22: Please, consider starting a new paragraph after '...then searched for in the subsequent images.'. ⇢ Page 3, Line 4: '...flow conditions. And Costa et al. (2000)...'. Extra point into the sentence. ⇢ Page 4, Line 12: 'Here, average water level and discharge are 48 cm and 2.2 $m^2$/s, respectively.'. Mean annual variables? ⇢ Page 4, Line 19: 'During this day discharge and water level were 5.7 $m^3$/s and 68 cm.'. Considering the information provided before (Average discharge and water levels are 6.9 $m^3$/s and 66 cm, respectively), why a decrease from 6.9 m3/s to 5.7 m3/s (17% of difference) is creating an increment from 66 cm to 68 cm in terms of water levels? ⇢ Page 5, Line 19: '...the performance of different cameras (fig. 1b). Two...'. Fig. 1a? ⇢ Page 6, Line 31: '...to matching failure. And if moving...'. Extra point into the sentence. ⇢ Page 9, Line 10: '...suitable at the Wesenitz. But at the Freiberger...'. Extra point into the sentence. ⇢ Table 2: What is s0? ⇢ Page 11, Line 29: '...2.7 $m^3$/s, which corresponds to the velocity measured by the ADCP...'. Discharge.

---

## Referee Comment (RC2) · Anonymous Referee #2 · 24 Jul 2019

This article presents a procedure to extract surface flow velocities and flow discharge from images captured either with permanent cameras or with drones in natural water systems. The methodology includes i) vibration removal from captured images, ii) feature identification with the GFTT algorithm, iii) feature tracking and trajectory development with normalized cross-correlation, iv) trajectory filtering based on a set of predetermined rules, and v) velocity estimation. Flow discharge can be estimated by reconstructing the bathymetry with structure-from-motion techniques and utilizing a velocity coefficient to estimate depth-averaged velocity.

The manuscript does not represent a substantial contribution to scientific progress by introducing new concepts, ideas, methods or data. Most of the algorithms applied in the procedure are well known and share similarities with existing literature (see,

for instance, Perks et al., 2016, Cao et al., 2018 and Tauro et al., 2018). Flow discharge measurements have already been demonstrated from Unmanned Aerial Vehicles (UAVs), see, for instance, Detert et al., 2017. Finally, several tools exist for flow calculation (see, for instance, Fudaa-LSPIV and RIVeR by Patalano et al.).

However, I acknowledge that a limitation to the implementation of image-based measurements is the lack of user-friendly and widely shared toolboxes. In this vein, the manuscript addresses the relevant scientific issue of establishing a procedure that can guide the users from image acquisition to flow discharge calculation. In this respect, the manuscript may be appreciable to the HESS readership as a technical note, and provided the focus of the article is targeted on the presentation of the tool and on its performance. Regarding the scientific quality and validity of the applied methods, many details are missing and, in its current form, it is difficult to evaluate the scientific soundness of the work. Additional experimental and analytical justification and, sometimes, data would be mandatory to establish a novel procedure. Finally, the presentation of the work should be improved as well as several figures.

In the following, I report major comments.

1. An important flaw of the work is that the computational tool is barely presented and recalled to during the manuscript. Since the focus of the paper is the introduction of a new procedure, the work should clearly state the underlying assumptions of the algorithms, required data and expected outputs. Some of these points are only mentioned briefly in the supplementary material and they are not given the right visibility. For instance, I believe it should be made clear that the water level is an input to the procedure, as well as a decent number of ground control points. The sentence "the provided velocity tracking tool allows for a contact-less measurement of spatially distributed velocity fields and to estimate river discharge in previously ungauged and unmeasured regions" should, therefore, be properly edited. Another important point regards the limitations of the procedure with respect to required inputs. For instance, it looks like images need to capture river banks in order for image co-registration to be

effective. This is a remarkable limitation and it should be clearly stated for users and readers.

2. Since the Authors claim that a new procedure is being introduced, a motivation on the selection of the specific sites should be provided. If the sites mostly differ in the morphology of their river bed, the bathymetry of both of them should have been independently (that is, not with images) measured and considered as a benchmark for structure-from-motion results.

3. Details on the ADCP benchmark measurements are missing. For instance, it is not clear how surface flow velocities were extrapolated from a range of 14 cm near the water surface. Given the rather shallow depth of both streams, it is surprising the Authors did not try to reconstruct the full velocity profile with the ADCP. Wind effects are not mentioned as well as alternative possible sources of noise in the data.

4. The description of the optical experimental setup is also unclear. The orientation angles of the optical axes of the cameras are not provided. Also, in case of experiments on the Wesenitz, even if three terrestrial cameras are installed along the cross-section, none of them captures the entire width of the stream. Using diverse optical parameters for the cameras could have been interesting if results had been better discussed and referred to such settings.

5. Most of the presented algorithms share common traits with already published material. However, some of them introduce novel aspects whose accuracy is not adequately assessed in the manuscript. Was the co-registration tested elsewhere before? Was it tested in windy conditions, under different camera orientations/frequencies/resolutions? What about the feature search area and pose estimation? What are the parameters such procedure is sensitive to? Was it validated in diverse conditions? If the method was only tested in the two case studies reported in the paper, then how can this tool be regarded as a robust alternative to thoroughly tested and used ones?

6. Some of the velocimetry phases require the definition of threshold values. It is not clear if they can be edited based on the specific case study. Even if this is possible, I believe the Authors should provide some guidance for the selection of appropriate values. For instance, what are nearest neighbor area dimensions that allow to find strong clusters of particles? Or which is a suitable number of particles? I believe such parameters are highly dependent on the specific experimental conditions, and automatic ways of computing them may be developed rather than asking for an intensive visual inspection of images by the users. Similarly, are search area dimensions pre-defined or inputs to the workflow? Introducing search area dimensions automatically poses constraints on the admissible frequencies and, therefore, flow velocities to be observed. In the track filtering, the criterion of the minimum number of frames across which the features have to be traceable also causes a constraint on measurable flow velocities and camera frequencies. Again the users should be aware of these implications and guided towards a sound selection.

7. The velocimetry procedure involved multiple filtering of particles and trajectories. This may be inefficient as compared to alternative approaches that perform the filtering only once. However, nothing is mentioned on the efficiency of the procedure. What are computational times related to image frequency and resolution? In several instances the Authors recommend to capture adequately long videos. Nonetheless, this can be time consuming and introduce additional variability due, for instance, to the occurrence of unevenly spaced tracers.

8. Transformation of trajectories to rasterized cells is not clear.

9. How was the velocity coefficient estimated? This is generally an approximate methodology that is not adequate in case of irregular sections. Since water level is an input to the procedure and the bathymetry of the stream reach is reconstructed, why weren't alternative approaches be considered and integrated for flow discharge computation?

[Figure]

10. Figures should be improved. For instance, in Fig.1 panels a and b are misplaced. Also, it would be nice, for each case study, to overlap the field of view captured by each camera to facilitate velocity comparison (same difficulty in Fig.9). In Fig.3a, all points are colored, it is unclear what the Authors are referring to. In Figs. 3 and 4 it would be nice to see the influence of the various steps of the filtering. In Fig.7, points that are far from the center of images do not necessarily display higher standard deviation. This should be commented and motivated in the manuscript.

11. It would be nice to see the tracks that fall within 1 m from the ADCP measurements in a figure. In some cases the computation is done on a very different number of trajectories regardless of the cluster-based filtering. Were values in Table 3 weighed by the number of track counts?

12. Even if the manuscript is mostly well readable, several typos and sentences should be improved. Some units are wrong. The sentence at lines 4 to 6 on page 7 is unclear.

---

## Author Response (AR1)

First of all, we would like to thank both referees (Salvatore Manfreda and one anonymous referee) that they invested their time and reviewed our manuscript in great detail to provide very constructive comments. We gladly considered each note to improve our submission.

**Salvatore Manfreda (Referee)**

salvatore.manfreda@unibas.it

The main contribution of the present manuscript is oriented in exploring surface flow velocities and discharge estimations using fixed cameras and UASs devices. A full and automatic workflow is introduced for the estimation of
the variables mentioned above. Two case studies are considered for validation purposes, namely the Wesenitz (paved) and Freiberger Mulde (natural). ADCP data were collected for benchmarking purposes. The manuscript is almost well written and easily understandable. Its length is also appropriate.

*Major comments:*

Section 2.2.1 Reference data: The authors stated that surface flow velocities were extrapolated using ADCP measurements. However, they do not say anything about the process. Please, add information on the extrapolating process. Additionally, at Wesenitz case study, only one cross-section was measured. Why such a decision? (consider that for a rigours comparison between image-velocimetry results and reference velocities is better not to use only
local reference velocities).

- Thank you for your comment. We clarifed in the revised manuscript how the ADCP measurements were extrapolated. Extrapolation of surface flow velocities was performed by a procedure suggested by Adler (1993) and also described in Morgenschweis (2010). The procedure is implemented in the AGILA software;
thus we believe that further details were not required. In general, it approximates a power function to the measured vertical velocity profile for each ADCP ensemble individually. Then, surface velocity (vs) is calculated by:

$$vsi = ai * hi^{\wedge}(1/6)$$

with h – water depth and
a – factor (determined from measured depth-depended velocities) for each ADCP ensemble, with
i – number of the ensemble, representing the position within the cross section.

This means, that surface velocities were extrapolated using all velocity measurements of the ADCP. At the Wesenitz site, ADCP cell sizes of 3 centimetres were used, which resulted in up to 10 depth-depended velocity measurements per ensemble.

o Adler, M.: Messungen von Durchflüssen und Strömungsprofilen mit einem Ultraschall-Doppler-Gerät
(ADCP). Wasserwirtschaft (83) 1993, H. 4, S. 192–196.
o Morgenschweis, G.: Hydrometrie, Springer-Verlag Berlin Heidelberg, S. 582, 2010. DOI: 10.1007/978-3-642-05390-0_1

- The Wesenitz study site is located at the gauging station. It is a straight channel with paved, trapezoidal cross
sections and nearly uniform flow conditions. Thus, surface velocities vary only slightly and one cross section seems to be representative. We performed more measurements with different water depths and locations showing similar results. But we did not want to put so much focus on this because the idea of the manuscript is to compare measurements with different sensors and platforms under uniform and non-uniform flow conditions.

Section 2.2.2 Image-based data: The authors used a low-resolution camera for video acquisitions at the Freiburger Mulde case study. Justify such a decision considering that low-cost smartphones can reach a better resolution.

- Indeed, the authors used a low-resolution camera. The specific camera was originally chosen because it is
also able to record high speed videos. However, analysis of these videos revealed that high speed frame rates do not necessarily improve the tracking quality but increase processing time significantly. Thus, we focused on the video with the lower frame rate, although the image resolution was low. Nevertheless, comparing our results to the ADCP measurements could still reveal the high accuracy potential highlighting that even low-resolution cameras can be used for the task of flow velocity and discharge measurements. In addition, the
SLR cameras at the Wesenitz study site enabled a detailed analysis of image velocimetry with imagery with higher resolutions.

Section 2.4.3 Feature tracking: The authors stated, 'In this study, features are tracked for 20 frames and new features are detected every 15th frame'. Is there any reason for these numbers? Why did the authors decide a new detection
every 15 frames?

- The decision for tracking for 20 frames and detecting features every 15 frames was chosen after evaluating different choices for tracking and detection. Although, other choices were possible (e.g. 10 frames tracking and every 10 frames detection), we settled with these settings as the results revealed steadiness and
processing time was acceptable. This choice was therefore based on our experience with both rivers.
- The more frames are tracked across; the more reliable and robust tracking results are possible because the later filtering will have a larger sample for processing. However, the longer features are tracked the longer the processing time is going to be. Choosing feature detection every 15 frames allowed us to densify the final feature tracks. Features can change and new features enter the area of view although the already detected
features are still tracked. Thus, it can be suitable to detect features more frequently than the number of frames they are tracked across. Thank you for highlighting the lack of explanation in the manuscript. We added this information to the revised manuscript.

Section 2.4.4 Track filtering: This subsection is relevant and deserves a better explanation of the filtering criteria. For
example, it would be positive to add a figure showing an example of application of the different filtering criteria (e.g. what is the reference for the angles?).

- Thank you for your comment. In fig. 5 we already implemented four sub-figures, which show how the different filtering steps improve the tracking results. We added two more sub-figures to highlight more
specifically, how the consideration of sub-track directions and the deviation from the average flow direction improves overall velocity field reliability.
- The references for the angles are chosen differently. The average flow direction is calculated from all tracks and then a buffer value is chosen to exclude tracks that exceed the average flow direction by a specified threshold. This threshold has to be defined considering the general variability of the river surface flow
pattern. The other criteria concern the steadiness of the tracks. If the standard deviation of the sub-tracks is above a specific value, they are excluded because we assume a steady flow for a track. Again thresholds are chosen considering the general flow characteristic of the observed river. We added some more information to the revised manuscript regarding the choice of the thresholds.

Section 2.4.5 Velocity retrieval: The authors stated: 'The threshold is defined as the sum of the average velocity with a multiple of its standard deviation'. Please, add information about the 'multiplying factor' of the standard deviation.

- The multiplying factor has to be chosen according the quality of the filtering results. If the factor is set to a high value only a few values, which deviate strongly from the average velocity, are removed. And if a low value is chosen, many more tracks are filtered out, which might be wanted in situations, when solely the most reliable tracking results are aimed for. We added this information to the revised manuscript.

Section 2.4.5 Velocity retrieval: The authors stated: 'For a better visualisation, final flow velocity tracks are rasterized'. Please, add information about the block assumed for the rasterizing process. If the comparison of estimated velocities is made with the rasterized velocities, please mention it and discuss implications.

- In this study we assumed a block of 20 pixels. However, we did not compare the rasterized velocities. We only used the original velocity tracks for comparison to the reference to avoid inaccuracies due to interpolation errors. In this study, the rasterized data is only considered as a visualisation tool and therefore we removed it in the revised manuscript to avoid confusion. Instead, we implemented a figure that contains the final tracks and the location of the ADCP track including a buffer to identify, which features were used for velocity comparison.

Section 3.2 Flow velocity measurements at the Wesenitz: The authors stated 'However, it is difficult to perform exact comparison to the ADCP measurements because the precise location of the ADCP cross-section in the local coordinate system of the river reach is not known as the ADCP boat was not equipped with any positioning tool and its movement across the water surface was neither tracked nor synchronised. Therefore, the accuracy assessment of the spatial velocity pattern is limited'. This is a critical issue that may limit the validation of the procedure. Do you have any alternative strategy to quantify ADCP positions in order to allow a realistic comparison?

- In this study, we were able to identify the start and end points of the cross-sections in the imagery at the shore. Therefore, we could approximately estimate the locations of the cross-sections. However, the location could only be estimated in the dm-range, which allows for velocity comparison if the surface flow velocity pattern does not become too variable within shortest distances. We just wanted to highlight that this has to be kept in mind. We added this info to the revised manuscript.

*Minor comments:*

Page 2, Line 2: '. . .observe flash floods. And Le Coz et al. . .'. Please, remove the point before 'and'

- Thank you for the comment. We changed this in the revised version.

Page 2, Line 22: Please, consider starting a new paragraph after '. . .then searched for in the subsequent images.'.

- Thank you for the comment. We added a new paragraph in the revised version.

Page 3, Line 4: '. . .flow conditions. And Costa et al. (2000). . .'. Extra point into the sentence.

- Thank you for the comment. We removed the extra point in the revised version.

Page 4, Line 12: 'Here, average water level and discharge are 48 cm and 2.2 m 2 /s, respectively.' Mean annual variables?

- Indeed, these are annual averages. Thank you for highlighting this. We added this information in the revised version. However, please see the next comment for further details.

Page 4, Line 19: 'During this day discharge and water level were 5.7 m 3 /s and 68 cm. Considering the information provided before (Average discharge and water levels are 6.9 m 3 /s and 66 cm, respectively), why a decrease from 6.9 m3/s to 5.7 m3/s (17% of difference) is creating an increment from 66 cm to 68 cm in terms of water levels?

- We checked the numbers again. They are correct. The measured values at that day (5.7 $m^3$/s and 68 cm) were obtained by continuous water level measurement and application of the rating curve, which was valid during that time. However, average discharge and water levels are long-term averages based on more than 50 years of measurements. In comparison with the values of that day, we see two effects, which are responsible for the differences. First, rating curves are changing over time (At the moment, a discharge of 5.7$m^3$/s is assigned with a water level of 66 cm). Second, rating curves reveal a nonlinear behaviour. With increasing water level, the discharge increases stronger, which has an impact on the averages. Thus, direct comparison of both pairs of values is not possible. However, to avoid this confusion we compared the values of that day to the average values of the hydrological year 2016, which are 5,6 $m^3$/s discharge and 65 cm water level, which also changed for the Wesenitz case study to obtain consistency.

Page 5, Line 19: '. . .the performance of different cameras (fig. 1b). Two. . .'. Fig. 1a?

- Indeed, this would be figure 1a. Thank you for noticing. It was changed in the revised manuscript.

Page 6, Line 31: '. . .to matching failure. And if moving. . .'. Extra point into the sentence.

- Thank you for your comment. We corrected this in the revised version.

Page 9, Line 10: '. . .suitable at the Wesenitz. But at the Freiberger. . .'. Extra point into the sentence.

- Thank you for your comment. We corrected this in the revised version.

Table 2: What is s0?

- s0 is sigma 0 and it is a resulting quality parameter of the adjustment of the spatial resection. It provides information about how well observed values fit to the adjusted values. We added this information to the table heading.

Page 11, Line 29: '. . .2.7 m 3 /s, which corresponds to the velocity measured by the ADCP. . .'. Discharge.

- Thank you for noticing. We changed this to discharge in the revised version.

**Anonymous Referee #2**

This article presents a procedure to extract surface flow velocities and flow discharge from images captured either with permanent cameras or with drones in natural water systems. The methodology includes i) vibration removal from captured images, ii) feature identification with the GFTT algorithm, iii) feature tracking and trajectory development with normalized cross-correlation, iv) trajectory filtering based on a set of predetermined rules, and v) velocity estimation. Flow discharge can be estimated by reconstructing the bathymetry with structure-from-motion techniques and utilizing a velocity coefficient to estimate depth-averaged velocity.

The manuscript does not represent a substantial contribution to scientific progress by introducing new concepts, ideas, methods or data. Most of the algorithms applied in the procedure are well known and share similarities with existing literature (see, For instance, Perks et al., 2016, Cao et al., 2018 and Tauro et al., 2018). Flow discharge measurements have already been demonstrated from Unmanned Aerial Vehicles (UAVs), see, for instance, Detert et al., 2017. Finally, several tools exist for flow calculation (see, for instance, Fudaa-LSPIV and RIVeR by Patalano et al.). However, I acknowledge that a limitation to the implementation of image-based measurements is the lack of user-friendly and widely shared toolboxes. In this vein, the manuscript addresses the relevant scientific issue of establishing a procedure that can guide the users from image acquisition to flow discharge calculation. In this respect, the manuscript may be appreciable to the HESS readership as a technical note, and provided the focus of the article is targeted on the presentation of the tool and on its performance. Regarding the scientific quality and validity of the applied methods, many details are missing and, in its current form, it is difficult to evaluate the scientific soundness of the work. Additional experimental and analytical justification and, sometimes, data would be mandatory to establish a novel procedure. Finally, the presentation of the work should be improved as well as several figures.

- Thank you very much for you detailed notion on our manuscript. Indeed, we did not aim at presenting a completely new concept regarding image velocimetry, as we also referred to the literature in the manuscript. We rather focus on improving existing methods, e.g. such as so far missing automatic image co-registration approaches (including performance assessment) or automatic extraction of feature search areas considering the water surface and river topography to improve the robustness and processing time of feature tracking. We believe that these improvements are relevant scientific contributions as they are important for an increased flexible application also for non-experts in image velocimetry. However, in the revised manuscript we listed the already existing tools for completeness. Furthermore, although various literature regarding image velocimetry already exists, we believe, there is still a contribution missing that demonstrates the applicability of PTV from terrestrial as well as UAV perspectives revealing the flexibility of the method.
- We focus our study on introducing the velocimetry tool. Therefore, we chose two different study sites with uniform and non-uniform flow conditions to evaluate the chances and limits of our tool. Furthermore, although some usable tools already exist for image velocimetry, they are not all capable to perform the whole processing chain (image stabilization, automatic feature search area extraction, feature detection, tracking with PTV, filtering and scaling tracks) in one tool considering GCPs or direct camera position/orientation information for referencing. Due to the focus on the tool and data processing, we submitted our manuscript as a technical note, which the reviewer also acknowledges would be suitable and useful to the HESS reader.
- We present the tool and evaluate its performance with independent ADCP measurements at two sites. However, within the revised manuscript, we focused even stronger on the tool itself and how it can be used. And we explained further what the limits/constraints of our tool are, what pre-requisites/input are needed, and what results can be expected.
- Regarding the statement that details are missing to validate the quality and validity of our method, we are afraid that we would need a more detailed explanation what is missing. We do not believe that additional experiments or data is necessary as we already demonstrate with two different sites, revealing different flow and topographic characteristics, the suitability of our tool and discuss where the limits are (e.g. irregular profiles where no hydraulic modelling is considered to improve the estimation of the velocity coefficient). Furthermore, we do not aim to develop a completely new approach but rather improve and combine existing velocimetry tools, which is stated in the last paragraph of the introduction as well as in the added section 'Limits and perspectives'.

- In the revised form we are more specific regarding the tool, improved the readability, and we improved figures, where needed.

*In the following, I report major comments.*

1. An important flaw of the work is that the computational tool is barely presented and recalled to during the manuscript. Since the focus of the paper is the introduction of a new procedure, the work should clearly state the underlying assumptions of the algorithms, required data and expected outputs. Some of these points are only
mentioned briefly in the supplementary material and they are not given the right visibility. For instance, I believe it should be made clear that the water level is an input to the procedure, as well as a decent number of ground control points. The sentence "the provided velocity tracking tool allows for a contact-less measurement of spatially distributed velocity fields and to estimate river discharge in previously ungauged and unmeasured regions" should, therefore, be properly edited. Another important point regards the limitations of the procedure with respect to
required inputs. For instance, it looks like images need to capture river banks in order for image co-registration to be effective. This is a remarkable limitation and it should be clearly stated for users and readers.

- Thank you for this comment. In the revised manuscript we referred more closely to our tool. Furthermore, different processing stages (e.g. expected output, required data) are also explained in the tutorial of the tool,
which is in the supplemental material of the manuscript.
- If the camera position/orientation is known (e.g. from precise GNSS and IMU information), actually no GCPs are needed. This missing information is provided in the revised manuscript.
- We adjusted the sentence concerning the application in ungauged regions in the regard that of course some scaling information is needed.
- We already state and discuss in the manuscript that the shore line of the river has to be visible (p 7 l 2-3, p 10 l 1-3). However, we discussed some more the corresponding limit that very wide rivers could solely be observed from high flying altitudes (potentially reducing the visibility of features to track) or with cameras with wide opening angles (potentially resulting in stronger lens distortions, which however can be corrected with suitable calibration procedures).

2. Since the Authors claim that a new procedure is being introduced, a motivation on the selection of the specific sites should be provided. If the sites mostly differ in the morphology of their river bed, the bathymetry of both of them should have been independently (that is, not with images) measured and considered as a benchmark for structure-from-motion results.

- We would like to highlight that we do not claim that a new procedure is introduced. We state for instance that we rely in PTV and Shi-Tomasi feature detection. We solely claim that our approach is automatic in most regards and that it is independent from the acquisition perspective.
- Indeed, both study sites were chosen due to their morphological differences. However, the focus of our study
is not on SfM because we only consider it as a further tool. Therefore, we refer to a previous study by Dietrich (2017), who demonstrated the usability of SfM for bathymetry, instead of establishing another benchmark. In another study (Eltner et al, 2018) we were able to demonstrate the usability of SfM to measure the bathymetry (actually at the same gauge at the Wesenitz comparing SfM to a cross-section measured with total station resulting in deviation below). We added this information to the revised manuscript to further
acknowledge the suitability of our approach.

      o  Dietrich, J.T.: Bathymetric Structure-from-Motion: extracting shallow stream bathymetry from multi-view stereo photogrammetry. Earth Surface Processes and Landforms, 42(2), 355-364, 2017

o   Eltner, Anette, Elias, M., Sardemann, H., Spieler, D.: Automatic Image-Based Water Stage Measurement for Long- Term Observations in Ungauged Catchments. Water Resources Research, (54), WR023913, 2018

3. Details on the ADCP benchmark measurements are missing. For instance, it is not clear how surface flow velocities were extrapolated from a range of 14 cm near the water surface. Given the rather shallow depth of both streams, it is
surprising the Authors did not try to reconstruct the full velocity profile with the ADCP. Wind effects are not mentioned as well as alternative possible sources of noise in the data.

    -   Because Salvatore Manfreda made a similar comment regarding the ADCP measurements, we copied and pasted our answer to review 2:
-   We clarified in the revised manuscript how the ADCP measurements were extrapolated. Extrapolation of surface flow velocities was performed by a procedure suggested by Adler (1993) and also described in Morgenschweis (2010). The procedure is implemented in the AGILA software; thus we believed that further details were not required. In general, it approximates a power function to the measured vertical velocity profile for each ADCP ensemble individually. Then, surface velocity (vs) is calculated by:

$$vs_i = a_i * h_i^{1/6}$$

with h – water depth and
a – factor (determined from measured depth-depended velocities) for each ADCP ensemble, with
        i – number of the ensemble, representing the position within the cross section.

This means, that surface velocities were extrapolated using all velocity measurements of the ADCP. At the Wesenitz site, ADCP cell sizes of 3 centimetres were used, which resulted in up to 10 depth-depended velocity measurements per ensemble.

        o   Adler, M.: Messungen von Durchflüssen und Strömungsprofilen mit einem Ultraschall-Doppler-Gerät (ADCP). Wasserwirtschaft (83) 1993, H. 4, S. 192–196.
        o   Morgenschweis, G.: Hydrometrie, Springer-Verlag Berlin Heidelberg, S. 582, 2010. DOI: 10.1007/978-3-642-05390-0_1

    -   Thank you for your advice regarding sources of noise in the data. We added this in the discussion since this is also a general limitation of all measurement systems relying on surface velocities. During our measurement campaign, we had nearly windless conditions.

4. The description of the optical experimental setup is also unclear. The orientation angles of the optical axes of the cameras are not provided. Also, in case of experiments on the Wesenitz, even if three terrestrial cameras are installed along the cross-section, none of them captures the entire width of the stream. Using diverse optical parameters for the cameras could have been interesting if results had been better discussed and referred to such settings.

-   Thank you for your comment. In the revised manuscript, we provided an appendix that contains the information about the exterior orientation parameters (including the accuracy). We did not include it in the original manuscript because we did not see the relevance of further discussion of the orientation parameters. In contrast, we wanted to show that our approach is independent of the exterior orientation of the camera, as long as it can be determined (either by direct geo-referencing or indirect from control points).
-   We would like to note that all three terrestrial cameras at the Wesenitz capture the entire width of the stream. However, there might be a confusion if fig. 8 is considered to assess the stream coverage. Unfortunately, the right shore is not completely displayed, although it has been captured (e.g. see fig. 3, displayed for camera 1200D-I, and compare to fig. 8), because features were not detected at the right shore (due to missing floating particles and sun reflections) and therefore we overlaid the legend over the right shore to save some figure space. We added this notion to the revised manuscript.

5. Most of the presented algorithms share common traits with already published material. However, some of them introduce novel aspects whose accuracy is not adequately assessed in the manuscript. Was the co-registration tested elsewhere before? Was it tested in windy conditions, under different camera orientations/frequencies/resolutions? What about the feature search area and pose estimation? What are the parameters such procedure is sensitive to? Was it validated in diverse conditions? If the method was only tested in the two case studies reported in the paper, then how can this tool be regarded as a robust alternative to thoroughly tested and used ones?

- We would like to state that we think, we could show in our study the usability of the automatic co-registration approach by comparing it to chosen control points (fig. 7). Furthermore, the accuracy and suitability of co-registration in different conditions over longer periods of time has been tested and illustrated in Eltner et al. (2018). We added a reference to that study in the revised paper. We also tested the co-registration under conditions of strong changes of orientation with lower image capture frequencies (2 per second) at another site, not included in this manuscript, and co-registration was performing well. However, we would like to note that in general as long as the shore is visible and provides enough texture in the images, co-registration works as the feature detection, matching and application of a transformation has been shown in many areas of application, also outside the environmental sciences as it is a standard approach.
- The pose estimation accuracy has already been provided in table 2 in the original manuscript. The accuracy of pose estimation depends on the distribution and accuracy of GCP measurement in object and image space. We provided some more information regarding basic principles of photogrammetric measurement in the revised manuscript.
- We are afraid that we are not sure what thoroughly tested and used alternatives the reviewer refers to. We refer our approach to existing velocimetry tools. And we are not aware of extensive (published) testing of other velocimetry tools. Most studies also solely evaluate their results at selected sites (e.g. Perks et al., 2016, Tauro et al., 2018, Detert et al., 2017). Furthermore, we apply the already tested PTV and GFTT approach and do not claim to develop a new detection and tracking tool but rather improve and combine existing methods.

6. Some of the velocimetry phases require the definition of threshold values. It is not clear if they can be edited based on the specific case study. Even if this is possible, I believe the Authors should provide some guidance for the selection of appropriate values. For instance, what are nearest neighbor area dimensions that allow to find strong clusters of particles? Or which is a suitable number of particles? I believe such parameters are highly dependent on the specific experimental conditions, and automatic ways of computing them may be developed rather than asking for an intensive visual inspection of images by the users. Similarly, are search area dimensions pre-defined or inputs to the workflow? Introducing search area dimensions automatically poses constraints on the admissible frequencies and, therefore, flow velocities to be observed. In the track filtering, the criterion of the minimum number of frames across which the features have to be traceable also causes a constraint on measurable flow velocities and camera frequencies. Again the users should be aware of these implications and guided towards a sound selection.

- Thank you for the comment. Indeed, the thresholds need to be set corresponding to the characteristic of each field site. We mentioned this in the revised manuscript. However, each threshold can be set by the user within in the provided tool and we give a guidance in the corresponding tool manual how the change of the thresholds influences the data processing. As the revised manuscript focuses in general more strongly at the tool, we also added some more details regarding the threshold choices from the tutorial to the revised manuscript.

- Indeed, an automatic selection of thresholds should be developed. However, this is on-going research and not focus of our study. But of course this is the way forward for instance considering machine learning approaches and observations of the river flow during different seasons.
- The search areas are defined automatically within the workflow if a 3D model and the water level is provided. However, the search area can also be provided to the tool using a mask image. However, we are afraid that we do not understand how the search area definition influences the admissible frequencies. The search area is set once for one observation period (over several seconds assuming a stable water level, which would be needed anyway for reliable discharge calculation) and the image area within the mask is processed for all the frames afterwards, actually decreasing processing time as only the area of interest is processed.
- The minimum number of frames to track can be set to 0 and therefore the constraints regarding flow velocities and camera frequencies are obsolete. We clarified this in the revised manuscript to avoid confusion in this regard.

7. The velocimetry procedure involved multiple filtering of particles and trajectories. This may be inefficient as compared to alternative approaches that perform the filtering only once. However, nothing is mentioned on the efficiency of the procedure. What are computational times related to image frequency and resolution? In several instances the Authors recommend to capture adequately long videos. Nonetheless, this can be time consuming and introduce additional variability due, for instance, to the occurrence of unevenly spaced tracers.

- Thank you for your note. The filtering only happens during two stages, once when features are detected, which therefore happens frequently, but this filtering is very fast. And data is filtered a second time after all features where tracked, which takes longer in processing depending on the number of detected frames but therefore this filtering is only performed once (as in alternative approaches). We provided some more guidance regarding the computational time for filtering considering frame rate and image resolution in the revised manuscript.
- Indeed, taking longer videos takes more time. But we do not think if a video is captured for 3 seconds or 20 seconds is not such a big impediment in regard of consuming time. And considering increased temporal measurement, the increase in observation length is actually beneficial for the cases of unevenly spaced tracers because chances are increased that features will eventually flow across the entire cross-section if the video is captured long enough.

8. Transformation of trajectories to rasterized cells is not clear.

- Thank you for your comment. We do not use the rasterized data for analysis and reference comparison. It is solely applied to rasterize the data. We chose a grid with a cell size of 20 pixels and interpolated the trajectories into this grid. However, in the revised manuscript we removed this figure.

9. How was the velocity coefficient estimated? This is generally an approximate methodology that is not adequate in case of irregular sections. Since water level is an input to the procedure and the bathymetry of the stream reach is reconstructed, why weren't alternative approaches be considered and integrated for flow discharge computation?

- The velocity coefficients were estimated from the ADCP measurements (as mentioned on p9, line 9). To be more specific, the velocity coefficient was estimated dividing the mean velocity of the cross section with the average surface velocity. In our study, we used a method for discharge calculation, which is commonly applied for measurement systems relying on surface velocities. Furthermore, we aimed for a direct comparison of the ADCP measurements with the variables measurable by image-based approaches. However, we agree that alternative approaches like hydraulic approaches could be useful and worth to be investigated. But we believe that this is a separate topic. It is beyond the focus of this technical note and should be explored in more detail in further studies as research in this regard is still sparse. We addressed
this in the discussion.

10. Figures should be improved. For instance, in Fig.1 panels a and b are misplaced. Also, it would be nice, for each case study, to overlap the field of view captured by each camera to facilitate velocity comparison (same difficulty in Fig.9). In Fig.3a, all points are colored, it is unclear what the Authors are referring to. In Figs. 3 and 4 it would be nice
to see the influence of the various steps of the filtering. In Fig.7, points that are far from the center of images do not necessarily display higher standard deviation. This should be commented and motivated in the manuscript.

-    Regarding fig. 1, we would like to mention that the panels are indeed correct; a corresponds to the Wesenitz and b to the Freiberger Mulde.
-    In the revised manuscript we added the field of view of the cameras in figure 1 and the location of the ADCP measurements to figure 8 and 9 (7 and 8 in the revised manuscript).
-    In fig. 3a points above the water surface are coloured with RGB information from the imagery and points under the water surface are all plotted in white. To clarify which points are meant, we differentiated between 'colorized according to their object colour' and 'white points' in the revised caption.
-    Regarding the different processing step, we added further sub-figures to the show the results of the single steps of filtering.
-    We are afraid that we are not able to assess why points further from the centre should reveal higher standard deviations regarding the image co-registration.

11. It would be nice to see the tracks that fall within 1 m from the ADCP measurements in a figure. In some cases, the computation is done on a very different number of trajectories regardless of the cluster-based filtering. Were values in Table 3 weighed by the number of track counts?

-    We added this information in figure 7 and 8 in the revised manuscript to allow for a better assessment of the
data comparison. Different numbers of trajectories are compared because each camera provided a different number of measured tracks (table 2); also after cluster-based filtering, which is a local filter, because the original density of features was already different as each camera was observing the river from a different perspective and with different camera geometry. This is mentioned in the manuscript.
-    No the values were not weighted by track counts. The number of track counts represents the sample size of
the statistics.

12. Even if the manuscript is mostly well readable, several typos and sentences should be improved. Some units are wrong. The sentence at lines 4 to 6 on page 7 is unclear.

-    Thank you for your comment. We read and corrected the revised manuscript carefully to correct false units and remove typos.
-    We rephrased the sentence lines 4-6 the following: Since most feature detectors look for regions with high contrast, points of interest would be found on the land, where contrast is usually higher than on the water surface. Thus, the river area has to be detected in the images and defined as the search area for tracking.

Some hints regarding author's changes in the revised manuscript have already been given in the comments section.

Here, a summary of author's changes to the manuscript based on comments of both referees is given.

-    Focus stronger on the tool itself and its usage, thus be more specific regarding the tool (keep information about each processing stage not just in the tutorial).

-    In general, explain more detailed how setting of thresholds influence tracking result and what parameters should be considered depending on the flow characteristics at different sites.

-    More detailed explanation regarding the individual thresholds for track filtering.

-    More detailed explanation regarding the parameter choices for feature detection and tracking (i.e. detection every nth feature and tracking for n number of features).

-    More guidance regarding computational time for filtering considering frame rate and image resolution.

-    Explain in more detail limits/constraints (e.g. wind, shore visibility, …), pre-requisites/ needed input (e.g. camera orientation/position or GCPs, …), and expectable results of the tool.

-    Explanation of the limits for the image-based to ADCP based velocity comparison due to the accuracy of the position estimation of the ADCP.

-    Clarification of the threshold (multiplying factor) definition of the statistical outlier filter.

-    Clarification of the ADCP extrapolation.

-    Add in figure 1 image area extents of each camera.

-    Addition of sub-figures (in fig. 5) that show the results of the different feature track filtering steps.

-    Removal of the figures with the rasterized velocities (fig. 8 and 9) and instead add figure with the final tracks. In addition, add to this figure location of ADCP cross-section(s).

-    Improve the readability of the manuscript.

-    Include an appendix with information about accuracies for estimation of exterior camera geometry.

Furthermore, we had to correct the s0 values in Table 2 because in the Discussion paper we accidently used a false pixel size (from the still image format instead of the frame format) to calculate s0 (it is originally correctly given in mm in the software, but we transformed it to pixel values for better assessment).

-    Include an appendix with information about the chosen parameters within the FlowVelo tool to perform velocity estimations with all camera variations.

-    Provide the manual of the tool as a separate supplement (in addition to the huge supplemental zip-file)

[revised manuscript text omitted]

Appendix A1: Accuracy of the estimation of the exterior camera geometry via spatial resection

| | | | nbr observations (GCPs used) | s0 [mm] | X [mm] | Y [mm] | Z [mm] | omega [rad] | phi [rad] | kappa [rad] | pixel size [mm] |
|---|---|---|---|---|---|---|---|---|---|---|---|
| Freiberger Mulde | UAV camera | estimated value | 7 | 0.006 | 14993 | 18976 | 30033 | -0.011 | 0.013 | 2.483 | 0.012 |
| | | std dev estimated value | | | 85 | 78 | 31 | 0.002 | 0.002 | 0.001 | |
| | Casio EX-F1 | estimated value | 6 | 0.003 | 6990 | 94615 | 6632 | 1.066 | -0.446 | 2.995 | 0.011 |
| | | std dev estimated value | | | 18 | 10 | 15 | 0.001 | 0.001 | 0.001 | |
| Wesenitz | UAV camera | estimated value | 6 | 0.036 | 203874 | 196051 | 215766 | -0.264 | -0.022 | 0.149 | 0.012 |
| | | std dev estimated value | | | 172 | 274 | 162 | 0.015 | 0.009 | 0.003 | |
| | 500D | estimated value | 5 | 0.015 | 199731 | 188229 | 202939 | 0.537 | -1.126 | 2.115 | 0.017 |
| | | std dev estimated value | | | 27 | 66 | 39 | 0.012 | 0.003 | 0.011 | |
| | 1200D-I | estimated value | 5 | 0.037 | 199546 | 191415 | 202847 | 0.020 | -1.270 | 1.638 | 0.012 |
| | | std dev estimated value | | | 42 | 169 | 80 | 0.042 | 0.008 | 0.038 | |
| | 1200D-II | estimated value | 4 | 0.019 | 199464 | 189855 | 203006 | 0.276 | -0.989 | 1.918 | 0.012 |
| | | std dev estimated value | - | - | 41 | 127 | 73 | 0.015 | 0.008 | 0.014 | - |

Appendix A2: Parameter settings chosen for each image-based tracking example in this study.

| | UAV camera (Freiberger Mulde) | UAV camera (Wesenitz) | terrestrial camera (Casio-EX F1; Freiberger Mulde) | terrestrial cameras (500D, 1200D-I, 1200D-II; Wesenitz) |
|---|---|---|---|---|
| *parameters feature detection* | | | | |
| threshold minimum brightness [8 bit] | 120 | 130 | 170 | 135 |
| search radius neighbouring features [pixel] | 10 | 5 | 5 | 50 |
| maximum neighbours in radius [ ] | 10 | 5 | 5 | 10 |
| maximum total number detected features [ ] | 3000 | 3000 | 3000 | 3000 |
| sensitivity (minimum quality detected feature) [ ] | 0.005 | 0.002 | 0.002 | 0.002 |
| *parameters feature matching* | | | | |
| template width | 10 | 6 | 10 | 10 |
| template height | 10 | 6 | 10 | 10 |
| search area width | 15 | 9 | 15 | 15 |
| search area height | 15 | 9 | 15 | 15 |
| shift search area in x direction | 0 | -1 | 0 | 1 |
| shift search area in y direction | 1 | 0 | 1 | 0 |
| subpixel measurement | TRUE | TRUE | TRUE | TRUE |
| perform LSM | FALSE | FALSE | FALSE | FALSE |
| *parameters iterations* | | | | |
| detect features every nth frame | 15 | 15 | 15 | 15 |
| track features for nth number of frames | 20 | 20 | 20 | 20 |
| select every nth frame for tracking | 1 | 1 | 1 | 1 |
| *parameter scaling* | | | | |
| frame rate (per second) | 25 | 25 | 30 | 30 |
| *parameters track filtering* | | | | |
| minimum tracking distance [pixel] | 0.1 | 0.1 | 0.1 | 0.1 |
| maximum tracking distance [pixel] | 10 | 10 | 10 | 10 |
| minimum number of tracks (count) [ ] | 13 | 13 | 13 | 13 |
| steadiness [deg] | 30 | 30 | 30 | 30 |
| maximum range of directional change [deg] | 120 | 120 | 120 | 120 |
| maximum deviation from main flow direction [deg] | 30 | 20 | 30 | 20 |
| statistical outlier threshold [ ] | 1.5 | 1.5 | 1.5 | 1.5 |

---

## Author Response (AR2)

We would like to thank referee #2 and the editor for their comments to further improve the manuscript. Please, see below our responses in detail to each remark.

Comments to Editor:

The paper can be accepted subject to minor revisions, i.e. addressing all comments by the referee. In particular, the title should be changed into "Technical Note: Flow velocity and discharge measurement in rivers using terrestrial and UAV imagery".

Besides the changes we made to comply with the comments of referee #2, we further corrected typos, adapted the title, added the referees to the acknowledgements and corrected information about the funding. Furthermore, we added one more sentence (L 198-199) that highlights that we used data from a further campaign at the Freiberger Mulde to reconstruct the underwater area. Due to the high turbidity on the day we made the velocity measurements it was not possible to see the submerged area, but a dataset from a campaign six weeks before was well suitable. Unfortunately, we missed to add this short explanation in the first manuscript and we hope it is possible to still implement this information to avoid future confusion.

Comments to Referee #2:

In this second round of revisions, the Authors have considerably enhanced their work. Herein I report a few minor suggestions for further improvement.

1. It should be clearly stated in the title that the article is a Technical Note.

Thank you for your suggestion. We changed the title correspondingly.

2. Line 772: " subpixel peak location is "

Thank you for highlighting the typo. We corrected it.

3. "to detect features more frequently" could be explained better

Thank you for your comment. However, we would like to leave the sentence as it is because we believe we already explain already why higher detection frequency than tracking frame number might be needed in some scenarios (i.e. "because features can change their appearance and new features can enter the area of view although the already detected features are still tracked"). We think, the sentence explains it sufficiently and any further statement would be repetitive.

4. Please define the range of sub-track directions from a geometric point of view. The term range is not fully appropriate to define angles.

Thank you for your comment. We changed the sentence accordingly to be more precise: "range of orientation angels of sub-tracks"

5. Assumptions on flow characteristics for track filtering as well as the parameter for the statistics-based velocity filtering need to be included in Section 3.5

We added some further explanations to the manuscript to discuss the consideration of flow characteristics and statistical based filtering (l. 448-452).

6. The sentence "colorized according to their object colour" is not very clear.

[revised manuscript text omitted]